# LLaMA Decoder as Vision Transformer

## Abstract

Using the same architecture for text and image is important for AI standardization. Recent multimodal models use a decoder-only Transformer to generate text and an encoder-only Transformer to extract image features. Can images use exactly the same language architecture? To answer this question, we aim at a LLaMa decoder as vision Transformer (ViT) classifier in this paper. Specifically, we start our trajectory by "LLaMAfy" a standard ViT step-by-step, *i.e.*, feed-forward net, normalization layer, causal self-attention and positional embedding, and point out a key issue—attention collapse—that result in the failure to the network training. Motivated by this observation, we propose *post-sequence class token*, enabling causal self-attention to efficiently capture the entire image's information. To improve model optimization behavior and enhance performance, we then introduce a *soft mask* strategy to gradually transform the attention from bi-directional to causal mode. The tailored model, dubbed as *image LLaMA (iLLaMA)*, maintains high consistency with LLaMA architecture, while matching up well against ViT, achieving 75.1% ImageNet top-1 accuracy with only 5.7M parameters. Scaling the model to ∼310M and pre-training on ImageNet-21K further enhances the accuracy to 86.0%. Its causal self-attention boosts computational efficiency and learns complex representation by elevating attention map ranks. Extensive experiments demonstrate iLLaMA's reliable properties: shape-texture bias, calibration, quantization compatibility, ADE20K segmentation and CIFAR transfer learning. We hope our study can kindle fresh views to visual architectures in the era of LLMs and contributes to standardized AI models.

## 1 Introduction

Using the same architectures for both text and images is important for building standardized AI systems. If architectures dealing with both modalities can be fully aligned, we can economically develop one set of operators to implement both models, and acceleration and optimization algorithms (Kwon et al., 2023; Dao et al., 2022; Dao, 2024; Shah et al., 2024) designed for one modality (*e.g.*, text) can be seamlessly transferred to another (*e.g.*, image).

Recent multimodal models use decoder-only large language models (LLMs, *e.g.*, LLaMA (Touvron et al., 2023a)) to generate text, given their superior scaling capabilities and performance. For image feature extraction, however, encoder-only ViTs (Dosovitskiy et al., 2020) are still used. a natural question is: *can decoder-only language architectures be used to handle images?*

However, the answer to this question is not intuitive. First, decoder-only architectures take a causal mode attention to process 1D text tokens, while encoder-only counterparts use a bi-directional mode attention to process 2D image tokens. Such intrinsic difference may affect the effectiveness of LLaMA training on visual tasks. Second, besides the attention mode, several differences still exist in architectural design choices between LLaMA and ViT, *i.e.*, feed-forward network (SwiGLU (Shazeer, 2020) vs MLP), normalization layer (RMSNorm (Zhang & Sennrich, 2019) vs LayerNorm (Ba et al., 2016)), and positional embedding (rotary (Su et al., 2024) vs learnable).

In this paper, we move closer to answering this question by introducing a decoder-only vision Transformer—*image LLaMA (iLLaMA)*, which adapts LLaMA decoder to an image classifier, as shown in Figure 1. Our exploration roadmap starts with an empirical study: replacing the components of LLaMA, *i.e.*, SwiGLU, RMSNorm, causal self-attention, and RoPE into a standard ViT step-by-step, and learn useful lessons along this adaptation process. Importantly, we observe an *attention*

**Image LLaMA (iLLaMA)**    **iLLaMA Block**    **Causal Self-Attention**

| | 65 | 70 | 75 | 80 | 85 |
|---|---|---|---|---|---|
| ViT (Baseline) | 73.8 | | 81.5 | 3.451 | |
| MLP → SwiGLU | 74.3 | | 82.0 | 3.407 | |
| LN → RMSNorm | 74.5 | | 81.7 | 3.406 | |
| Bi-di. SA → Causal SA | Failed | | | | |
| + Post-Sequence [cls] | 71.9 | | 80.6 | 3.599 | |
| Learnable PE → RoPE | 72.6 | | 81.2 | 3.618 | |
| + Learnable PE | 73.2 | | 81.2 | 3.531 | |
| Data Augmentation | 74.3 | | 81.3 | 2.990 | |
| + Soft Mask | 75.0 | | 81.6 | 2.955 | |

Accuracy / Training Loss for Tiny Regime

Figure 1: *Left*: **iLLaMA architecture.** *Right*: **our design roadmap.** Colored and gray bars represent the results of the tiny and base models. The red line depicts the training loss of the tiny model. iLLaMA strives to process visual tokens using standard LLaMa components, *e.g.*, causal self-attention. The proposed *PS [cls]* and *soft mask* strategy help overcome training challenges.

*collapse* issue when using causal self-attention directly for image classification, *i.e.*, the training loss fails to converge due to the causal attention mode. Specifically, the causal mask restricts the class token from accessing the image's global information, thereby hindering the model optimization. To this end, we propose a *post-sequence class token (PS [cls])* technique, repositioning the class token to the end of image tokens (details in Section 3.1). As a result, causal mask can keep the attention score between the class token and others, allowing the model to optimize stably. Further, we propose a *soft mask* strategy—transforming bi-directional mode attention to a causal mode one during training (details in Section 3.2). Soft mask does not alter the causal self-attention during inference but improves the network training behavior. We also evaluate the advantages of the causal self-attention in reducing computational complexity and enhancing the attention map rank.

Equipped with the proposed *post-sequence class token* technique and *soft mask* strategy, the decoder-only iLLaMA using pure causal attention can achieve comparable or even better classification performance than its encoder-only counterparts (*i.e.*, ViT, VisionLLaMA (Chu et al., 2024)). Beyond ImageNet-1K classification (Deng et al., 2009), we also conduct a thorough evaluation of other key properties of iLLaMA, including calibration, shape-texture bias, quantization compatibility, ADE20K semantic segmentation (Zhou et al., 2019), and CIFAR transfer learning (Krizhevsky et al., 2009). Experimental results show that iLLaMA delivers favorable and reliable performance to the encoder-only ViT, while maintaining a pure decoder design, fully aligned with LLaMA. More importantly, a spectral analysis on the attention map shows that compared to bi-directional counterparts, causal self-attention has a higher rank (see Figure 4), which allows for learning complex image representation. Based on this results, please rest assured to use iLLaMA as a suitable alternative to ViT for visual feature extraction. We summarize the contribution of our work as follows:

- We investigate several designs of using LLaMA decoder as an image classifier and learn useful lessons along the adaptation way. iLLaMA fully aligns with LLaMA in architecture.
- We identify the *attention collapse* issue when applying causal mode attention, and thus introduce a *PS [cls]* technique and a *soft mask* strategy to respectively to address this issue and improve model training behavior.
- Extensie experiments on ImageNet, transfer learning, along with practical properties such as quantization compatibility, calibration, shape-texture bias demonstrate that iLLaMA can be safely used as an efficient and reliable ViT alternative for image feature extraction.

## 2 PRELIMINARIES

### 2.1 TRANSFORMER ENCODER AND DECODER.

We briefly summarize the encoder and decoder in Transformer (Vaswani et al., 2017). Both of them basically consist of attention module and a MLP module, each followed by a residual connection. *The key difference between them is the mask scheme in their self-attention.* Encoders use bi-directional self-attention, and decoders employ causal self-attention and cross-attention. However, the latter is typically omitted in decoder-only LLMs (Touvron et al., 2023a;b), we thus focus on comparing causal and bi-directional attention as follows, in terms of the *mask* setting. Denote

$\mathbf{X} \in \mathbb{R}^{N \times d}, \mathbf{O} \in \mathbb{R}^{N \times d}$ as the input and output sequences, where $N$ and $d$ are sequence length and hidden dimension. $W_{\mathbf{q}}, W_{\mathbf{k}}, W_{\mathbf{v}} \in \mathbb{R}^{d \times d}$ denotes the linear mapping of query, key and value. Generally, self-attention can be formulated as (set head number and batch size as 1 for simplicity):

$$\mathbf{A} = \frac{1}{\sqrt{d}}(W_{\mathbf{q}}(\mathbf{X}) \cdot W_{\mathbf{k}}(\mathbf{X})^{\top}), \quad \mathbf{O} = \mathrm{Softmax}(\mathbf{A} + \mathbf{M}) \cdot W_{\mathbf{v}}(\mathbf{X}), \quad \mathbf{P}_{i,j} = 0, \quad \mathbf{Q}_{i,j} = \begin{cases} 0, i \geq j \\ -\infty, i < j \end{cases} \quad (1)$$

where $i, j \in [1, N]$, $\mathbf{A} \in \mathbb{R}^{N \times N}$, $\mathbf{M} \in \mathbb{R}^{N \times N}$ denote the attention map and mask. $\mathbf{P} \in \mathbb{R}^{N \times N}$, $\mathbf{Q} \in \mathbb{R}^{N \times N}$ are masks in the encoder and decoder, respectively. For a causal self-attention, we have $\mathbf{M} = \mathbf{Q}$. Such design allows subsequent tokens only attend to the preceding ones, but not vice versa. For a bi-directional self-attention, we have $\mathbf{M} = \mathbf{P}$, ensuring mutual visibility for each token.

## 2.2 RECENT LLMs-RELATED IMAGE MODELS.

Recent image models (Bai et al., 2023; Guo et al., 2024; El-Nouby et al., 2024) are trained with an autoregressive objective, targeting at solving visual tasks. Pang et al. (Pang et al., 2023) add a text pre-trained frozen LLM block to a ViT encoder to facilitate the performance. Our work, on the other hand, is motivated to explore in-depth how the decoder design in LLMs can be adapted to image models using simple supervised learning to achieve an architectural alignment. A concurrent work VisionLLaMA (Chu et al., 2024) proposes vision models based on the LLaMA components. Differently, we: 1) introduce causal mode attention from LLaMA, addressing the associated attention collapse issue, while VisionLLaMA retains an encoder architecture; 2) develop a soft mask technique to assist training the decoder; 3) expand the dataset to the larger ImageNet-21K to demonstrate scalability, achieving 86.0% ImageNet accuracy that outperforms VisionLLaMA's best results. Block details of ViT, VisionLLaMA, and our iLLaMA are compared in Appendix A.

## 3 "LLaMAFY" A STANDARD ViT: A ROADMAP

In this paper, we aim at a LLaMA decoder as a vision Transformer classifier. To this end, we first conduct an empirical study of gradually aligning LLaMA designs to a standard ViT step-by-step in Section 3.1. These designs include 1) feed-foward network, 2) normalization layer, 3) causal self-attention, 4) positional embedding. Further, we study training techniques to facilitate optimization in Section 3.2. Finally, in Section 3.3, we provide an analysis in terms of efficiency and attention map rank. We use ViT-T/16 and ViT-B/16 with around 5.7M and 86.4M parameters. We conduct experiments on ImageNet-1K (Deng et al., 2009), following the training recipe adopted from (Liu et al., 2023) (details in Appendix C.1). Considering the differences between visual perception and text generation tasks, we maintain ViT's non-autoregressive manner in our network. Each step change and the corresponding results are reported in Appendix D.

### 3.1 POST-SEQUENCE CLASS TOKEN: ACHIEVING ARCHITECTURE ALIGNMENT

**Feed-forward network (FFN)** module is implemented as multi-layer perceptron (MLP) in ViT and SwiGLU (Shazeer, 2020) in LLaMA. MLP consists of two sequential linear mappings. Meanwhile, SwiGLU combines three linear mappings, allowing for the modulation of high-dimensional features. We substitute the Transformer's MLPs with SwiGLUs, while maintaining comparable computational cost. As shown in Figure 1, this improves performance from 73.8% to 74.3%, and from 81.3% to 82.0% for the ViT-T/16 and ViT-B/16 regime. This highlights SwiGLU's effectiveness not only in language models but also in vision, inspiring further exploration of other components. *We will now use SwiGLU to substitute MLP in each block.*

**Normalization layer** is the key module in Transformers for stable training *i.e.*, layer normalization (LN) (Ba et al., 2016) in ViT and root mean square layer normalization (RMSNorm) (Zhang & Sennrich, 2019) in LLaMA. We replace all LNs with RMSNorms in our network and empirically observed that the accuracy of the ViT-T/16 regime increased from 74.3% to 74.5%. However, similar improvements in precision were not observed in the ViT-B/16 regime (from 82.0% to 81.7%). Nonetheless, compared to LN, RMSNorm removes the shift term computation, bringing simplicity to the network. *We will use RMSNorm instead of LN as the normalization layer in each block.*

**Causal mode attention leads to attention collapse issue.** The key component for causal mode attention in Transformer decoders is the causal mask, *i.e.*, a lower triangular mask matrix, illustrated

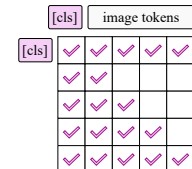

(a) causal mask          (b) causal mask w/ *PS [cls]* (ours)          (c) modified causal mask (ablation)

Figure 2: **Mask schemes**. (a) causal self-attention. (b) causal self-attention with our post-sequence class token (*PS [cls]*) method. (c) modified causal mask. Their ablation results are shown in Table 1.

in Eq. 1 and Figure 2(a). With such, each token can get the attention score of all its previous ones. We add the causal mask to our network via a non-autoregressive way. The reason is that visual perception tasks, unlike text generation, require only inference once. As a result, we observe that the training loss fails to converge in both ViT-T/16 and ViT-B/16 regimes (line 1 in Table 1). We posit that such issue stems from the influence of the lower triangular matrix, which prevents the class token from "seeing" other image tokens. As illustrated in Figure 2(a), when the class token is positioned at the start of the patch embedding, its attention score for all other image tokens gets zero due to a causal mask. We term this as the *attention collapse* issue, which leads to a loss of connection between the class token and other image patches, thereby hindering network optimization.

**Post-sequence class token (*PS [cls]*).** The attention collapse issue stems from the inappropriate placement of the token. To this end, we suggest a *PS [cls]* technique, by placing it at the end of the token sequence, without changing the causal mask, as shown in Figure 1 and 2(b). Such modification ensures that the class token can achieve global information about all image

Table 1: **Results of *PS [cls]* and the modified causal mask.** Training converges in both settings.

| Model | Tiny | Train Loss | Base | Train Loss |
|---|---|---|---|---|
| None | 0.1 | Failed | 0.1 | Failed |
| PS [cls] | 71.9 | 3.599 | 80.6 | 2.869 |
| Modified | 72.5 | 3.550 | 80.4 | 2.857 |

tokens, while maintaining a causal self-attention property. As a result, we observe that the attention collapse issue is eliminated and the training process starts to stabilize, leading the network performance to 71.9% for ViT-T/16 and 80.6% for ViT-B/16 regime, respectively (line 2 in Table 1).

To test our hypothesis about the reason of the attention collapse issue, we also explore a mask setting in Figure 2(c). In this setting, we do not change the position of the class token. Instead, we unmask the first row of the mask (*i.e.*, attention score of the class token) on the basis of the causal self-attention, termed as "modified causal mask". Ablation results (line 3 in Table 1) shows that both settings can solve the attention collapse issue as expected, and the "modified causal mask" leads to a better 72.5% accuracy for ViT-T/16 regime, validating our hypothesis about the reason. Although the results do not surpass the performance of bi-directional counterpart, they demonstrate the potential for optimizing causal mode attention for decoder-only image models. *We will employ causal self-attention with **the proposed PS [cls] method** in each block.*

**Positional embedding.** ViT use learnable positional embedding (LPE), typically adding it directly to the patch embedding. Meanwhile, rotary positional embedding (RoPE) (Su et al., 2024) is generally applied in LLMs (Touvron et al., 2023a;b), which functions in the attention of each block. We first use RoPE alone, which boosts the accuracy of ViT-T/16 and ViT-B/16 regimes to 72.6% and 81.2%, from 71.9% and 80.6%, respectively. The encouraging results illustrate that the concepts of "position" in image and text do not exist an inherent gap. Since LPE functions only once before all Transformer blocks, keeping it does not disrupt the alignment with LLaMA within each block. Thus, we reintroduce the LPE, which improves the accuracy of ViT-T/16 regime from 72.6% to 73.2%, suggesting that the two positional embeddings are not redundant but rather synergistic, enhancing network performance. *We will use both LPE and RoPE for positional embedding.*

*So far, we have studied the adaptation of LLaMA decoder as an image classifier, and as a result, we have settled on a final architecture dubbed iLLaMA. Next, we explore improved training strategies.*

## 3.2 Soft Mask: Improving Training Behavior

**Data augmentation.** Mixup (Zhang et al., 2018) and cutmix (Yun et al., 2019) that we used to train our iLLaMA (0.8 and 1.0), are borrowed from DeiT (Touvron et al., 2021)'s recipe. Unlike the bi-directional self-attention used in DeiT, causal self-attention affects the connection between

(a) soft mask scheme

(b) training curves w/ or w/o soft mask

Figure 3: **Soft mask.** (a) Gradually transitions from a bi-directional mask into a causal mask during training through a constant or linear schedule. (b) Ablation results of training loss and test accuracy.

image tokens. Meanwhile, these two hyper-parameters affect the content of the input image, which further influences the subsequent embedding. Thus, we reevaluate their impact on iLLaMA optimization. Specifically, we discover that a combination of 0.1 mixup and 0.1 cutmix improves the performance of the iLLaMA-T/16 from 73.2% to 74.3%, whereas a combination of 0.95 and 1.0 leads the iLLaMA-B/16 to a 81.3% accuracy. Other ablations are detailed in Section 4.1.

**Soft mask.** When observing objects, humans tend to perceive broad connections, then focus on specific details. Motivated by this, we propose a *soft mask* strategy to improve the model's training behavior—*starting with bi-directional mode attentions in the early training epochs and gradually shifting completely to causal mode attentions as the optimization goes*. Self-attention using soft mask can be formulated as:

$$\mathbf{A} = \frac{1}{\sqrt{d}}(W_{\mathbf{q}}(\mathbf{X}) \cdot W_{\mathbf{k}}(\mathbf{X})^{\top}), \quad \mathbf{O} = (\text{Softmax}(\mathbf{A}) \odot \mathbf{S}) \cdot W_{\mathbf{v}}(\mathbf{X}),$$

$$\mathbf{S} = \alpha\mathbf{B} + (1-\alpha)\mathbf{C}, \quad \mathbf{B}_{i,j} = 1, \quad \mathbf{C}_{i,j} = \begin{cases} 1, i \geq j \\ 0, i < j \end{cases} \tag{2}$$

where $i, j \in [1, N]$, $\mathbf{S} \in \mathbb{R}^{N \times N}$ denotes the soft mask, which is defined as a linear combination of a bi-directional mask $\mathbf{B}$ and a causal mask $\mathbf{C}$. $\alpha$ is the hyper-parameter controlling the mask configuration, *i.e.*, soft mask degenerates into $\mathbf{B}$ or $\mathbf{C}$ when $\alpha = 1$ or $\alpha = 0$, respectively. As illustrated in Figure 3(a), $\alpha$ involves three related hyper-parameters: 1) scheme: how $\alpha$ drops from 1 to 0: we try a linear or a constant scheme. 2) cutoff epochs: when will $\alpha$ drops to 0. 3) learning rate (lr) warmup (He et al., 2016; Goyal et al., 2017): this hyper-parameter overlaps with the duration of soft mask. We initially set the lr warmup epochs at 50, consistent with previous settings. When using a linear scheme with 50 and 25 cutoff epochs, we observe an improvement in performance for both iLLaMA-T/16 and iLLaMA-B/16 models, reaching 74.9% and 81.6% from 74.3% and 81.3%, respectively. Ablations results are detailed in Section. 4.1. To intuitively observe the impact of soft mask, we plot the training curve of the iLLaMA-T/16 in Figure 3(b), using a constant scheme with 50 cutoff epochs. When soft mask ends, we observe that although there was a sharp drop in accuracy, the model ends up achieving better performance. Similar case of the iLLaMA-B/16 are shown in Appendix F. Additionally, we discover that a lower learning rate warmup helps iLLaMA-T/16 achieve 75.0% top-1 accuracy, by using a constant scheme with 50 cutoff epochs. Therefore, we use this warmup method for iLLaMA-T/16. Notably with soft mask, the final training loss within both iLLaMA-T/16 and iLLaMA-B/16 decreases, suggesting an alleviation of potential underfitting.

### 3.3 ANALYSIS OF CAUSAL MODE SELF-ATTENTION

Finally, we analyze the advantages of using causal mode attention in iLLaMA, in terms of computational efficiency and image representation quality through the lens of attention map rank.

**Computational complexity.** For a self-attention with a sequence length $N$ and hidden dimension $D$, FLOPs are reported in Table 2 (RoPE is not involved as only attention related computations are calculated). Causal mode

Table 2: **Computational complexity.** Causal mask slightly reduces FLOPs required in the self-attention.

| Mode | Bi-directional | Causal |
|---|---|---|
| FLOPs | $4ND^2 + 2N^2D$ | $4ND^2 + N^2D + (\lfloor N^2/2 \rfloor + 1)D$ |

self-attention, due to lower triangular property of its attention map, slightly reduces the FLOPs compared to the bi-directional baseline—the degree of reduction grows as the sequence length increases.

**Attention map rank.** We examine the representation learning power of causal attention through a spectrum analysis. Following (Wang et al., 2020; Shu et al., 2021), we perform singular value decomposition on the attention maps of the pre-trained ViT-T/16 and iLLaMA-T/16 models. Next, we sort the singular values and plot a curve illustrating the relationship between the cumulative normalized singular values and matrix indices. The results are conducted using 30 images randomly selected from the ImageNet-1K validation set. As shown in Figure 4, the curve of ViT exhibits concave function characteristics, while the curve of iLLaMA is close to a linear function, indicating a more uniform distribution of singular values in iLLaMA's attention map. Approximating the matrix rank by the index at which the cumulative normalized singular value reaches 0.8, we observe that the index value of iLLaMA is about 48 higher than that of ViT (∼129-th v.s. ∼81-th). Under such premise, compared to ViT, the attention map of iLLaMA can be approximated with a certain error by a higher-rank matrix. Accordingly, the rank of the attention map may affect the expressive capabilities of the learned representations (Dong et al., 2021), suggesting that the causal self-attention in iLLaMA has the potential to learn complex visual representations, as demonstrated in Section 4.2. Detailed results are provided in Appendix E.

So far, we have finished the exploration process of iLLaMA with architecture alignment and improved training strategy. As a decoder-only Transformer, iLLaMA shows advantages in computational complexity and attention map rank via its causal mode attention. Notably, while all components of iLLaMA are essentially derived from LLaMA, only relying on them is insufficient for an effective training, as demonstrated in Section 3.3. In fact, the proposed *PS [cls]* and soft mask strategy effectively address this issue and assist in iLLaMA training. However, to achieve a comprehensive understanding of iLLaMA's properties, some useful evaluation should be conducted: 1) Scalability for large model capacities (>300M parameters) and dataset sizes (>10M training images, *e.g.*, ImageNet-21K). 2) Other practical evaluation dimensions, such as model calibration, shape-texture bias, downstream task performance, quantization compatibility, discussed below.

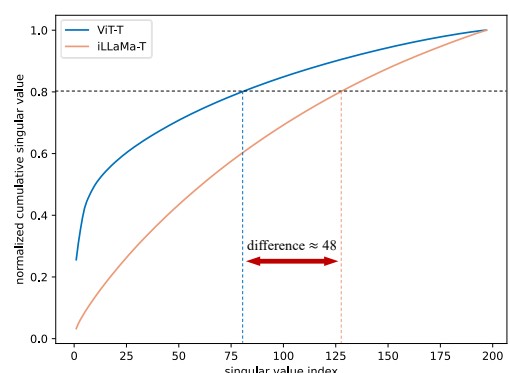

Figure 4: **Rank analysis.** Results of the attention map in head 1, layer 1 of the pretrained ViT-T and iLLaMA-T with $N = 197$. Difference between them is about 48.

## 4  EXPERIMENTS

In this section, we provide a comprehensive evaluation of iLLaMA. We first report ablation results, *e.g.*, the effectiveness of data augmentation and different soft mask strategies. Next, we compare iLLaMA with other strong baselines on ImageNet classification. Beyond ImageNet accuracy, we also examine its efficacy on calibration, shape-texture bias, and evaluate its compatibility with quantization-aware training and downstream task performance.

### 4.1  ABLATION STUDY

**Influence of data augmentation.** Base on the observation in Section 3.2, we examined multiple sets of cutmix and mixup settings, as reported in Table 5. We empirically observe that the smaller iLLaMA-T/16 are more sensitive to two data augmentation strategies and perform better with lower hyper-parameters, whereas the larger iLLaMA-B/16 are suited to higher ones. This may be related to the architectural differences between LLaMA's Transformer decoder and ViT's encoder type.

**Influence of soft mask scheduling strategies and epochs.** As mentioned in Section 3.2, the proposed soft mask technique includes three hyper-parameters, *i.e.*, schedule, cutoff epochs and lr warmup epochs. Here we evaluate the robustness of soft mask to hyper-parameter settings, with results detailed in Table 3. Beyond the *linear* schedule, inspired by (Liu et al., 2023), we also implemented a *constant* option. Additionally, we fixed the learning rate warm-up epochs at 50 and experimented with different cutoff epochs. The results reveal that the soft mask facilitates the optimization of iLLaMA under both linear and constant scheduling, suitable for models of both tiny and

Table 3: **Soft mask scheduling.** Results of tiny and base models on ImageNet-1K.

| Schedule | Cutoff Epochs | Tiny | Base |
|---|---|---|---|
| no softmask | - | 74.3 | 81.3 |
| linear | 25 | 74.8 | **81.6** |
| linear | 50 | **74.9** | 81.5 |
| linear | 100 | **74.9** | 81.5 |
| constant | 25 | 74.7 | 81.5 |
| constant | 50 | 74.8 | 81.5 |

Table 4: **Soft mask for training loss and testing loss.** Soft mask lowers both training and testing loss in tiny and base models, counteracting underfitting issue and thus leading to a better optimization.

| Model | Training Loss | Testing Loss |
|---|---|---|
| tiny | 2.990 | 1.121 |
| + soft mask | 2.955 ($\downarrow 0.045$) | 1.092 ($\downarrow 0.029$) |
| base | 2.868 | 0.843 |
| + soft mask | 2.828 ($\downarrow 0.040$) | 0.831 ($\downarrow 0.012$) |

base sizes. Moreover, setting the cutoff epochs to span a wide range from 25 to 100 is advantageous. Notably, the soft mask can be easily integrated into existing code frameworks (*e.g.*, timm (Wightman, 2019)) with negligible additional training costs, thereby facilitating future application.

**Influence of soft mask for training and testing loss.** Deep neural networks often face underfitting, marked by difficulty in continuously reducing training loss and resulting in poor test accuracy (Liu et al., 2023). We compare the training and testing losses of the iLLaMA-T/16 and iLLaMA-B/16 models with and without the use of the soft mask strategy. As shown in Table 4, soft mask can reduce training loss in both regimes, mitigating potential underfitting issue.

Table 5: **Mixup and cutmix ablation.** Results for tiny and base models.

| Mixup | Cutmix | Tiny | Mixup | Cutmix | Base |
|---|---|---|---|---|---|
| 0.8 | 1.0 | 73.2 | 0.8 | 1.0 | 81.2 |
| 0.5 | 0.4 | 73.8 | 0.9 | 0.9 | 81.2 |
| 0.3 | 0.3 | 73.9 | 0.9 | 1.0 | 81.2 |
| 0.2 | 0.2 | **74.3** | 1.0 | 1.0 | 81.2 |
| 0.1 | 0.1 | **74.3** | 0.95 | 1.0 | **81.3** |

## 4.2 IMAGENET-1K CLASSIFICATION

We conducted experiments on the ImageNet-1K Deng et al. (2009) benchmark with different model sizes (*i.e.*, iLLaMA-T/S/B/L). Detailed architecture configurations are shown in Appendix A. Our ImageNet-1K/21K (pre-)training and ImageNet-1K fine-tuning recipes are shown in Appendix C. We also study the use of LLaMA2-7B pre-trained weights for iLLaMA initialization, and the results are detailed in Appendix I.

**ImageNet-1K training.** We train iLLaMA-T/S/B on ImageNet-1K for 300 epochs with AdamW optimizer (Loshchilov & Hutter, 2019) and a batch size of 4096. The ImageNet-1K trained iLLaMA-T/B are, in fact, the outcome of the explorations completed in Section 3.2. For the settings of soft mask schedule, cutoff epochs, and learning rate warmup epochs, we tune slightly for the iLLaMA-S.

**ImageNet-21K pre-training.** We use the "Winter21 variant of ImageNet-21K-P" (refered to as ImageNet-21K) dataset (Ridnik et al., 2021) [1] for the large-scale pre-training of our iLLaMA, which contains 11,060,223 training images and 522,500 testing images from 10,450 classes. Only the training set was used. We pre-train iLLaMA-B/L on ImageNet-21K for 90 epochs using a constant soft mask schedule, with cutoff epochs and learning rate warmup epochs set to 30 and 5, respectively.

**ImageNet-1K fine-tuning.** For iLLaMA-B model trained on ImageNet-1K, we fine-tune at a resolution of 384×384. Similarly, for the iLLaMA-B/L model trained on ImageNet-21K, we fine-tune at resolutions of 224×224 and 384×384, respectively. All fine-tuning was conducted for 30 epochs using the AdamW optimizer. We follow DeiT (Touvron et al., 2021) for interpolating positional embeddings to allow our iLLaMA to handle inputs at a higher resolution.

**Results.** Table 6 shows a comparison between iLLaMA and other strong baselines, including ConvNets (ConvNeXt (Liu et al., 2022), ConvNeXt-V2 (Woo et al., 2023)), Transformers (ViT (Dosovitskiy et al., 2020), Swin Transformer (Liu et al., 2021)), MLPs (PoolFormer (Yu et al., 2022), VanillaNet (Chen et al., 2023)), and language model related models (AIM (El-Nouby et al., 2024), ViM (Zhu et al., 2024), VMamba (Liu et al., 2024), ViL (Alkin et al., 2024), and VisionLLaMA (Chu et al., 2024)). We present three observations: 1) The performance-parameter trade-off of iLLaMA surpasses some LM-related models (*e.g.*, AIM), presumably due to the causal attention and soft mask strategy. 2) iLLaMA exhibits a superior accuracy-throughput trade-off compared to strong hierarchical baselines such as ConvNeXt-V2-N/T/B. We attribute this to iLLaMA's isotropic design (each intermediate block has the same feature resolution), which benefits from a straightforward and efficient architecture, enhancing inference speed. 3) Scalability of model capacity and dataset

---

[1] downloaded from: https://www.image-net.org/download-images.php

Table 6: **ImageNet-1K accuracy.** Throughput (images/s) are tested on Nvidia A100 GPU with 1024 batch size. Hie.: Hierarchical, Iso.: Isotropic, Sup.: Supervised (pre-)training, AR.: Autoregressive pre-training. ♠ ConvNet, ■ Vision Transformer, ♣ MLP, ⌖ LM-related model, ★ LLaMA.

| Model | Dataset Used | Objective | Type | Image Size | Params | MACs | Throughput | Acc |
|---|---|---|---|---|---|---|---|---|
| ♠ ConvNeXt-S | IN-1K | Sup. | Hie. | 224×224 | 50M | 8.7G | 1185 | 83.1 |
| ♠ ConvNeXt-B | IN-1K | Sup. | Hie. | 224×224 | 89M | 15.4G | 877 | 83.8 |
| ♠ ConvNeXt-L | IN-1K | Sup. | Hie. | 224×224 | 198M | 34.4G | 543 | 84.3 |
| ♠ ConvNeXtV2-N | IN-1K | Sup. | Hie. | 224×224 | 15.6M | 2.45G | 2120 | 81.2 |
| ♠ ConvNeXtV2-T | IN-1K | Sup. | Hie. | 224×224 | 28.6M | 4.47G | 1362 | 82.5 |
| ♠ ConvNeXtV2-B | IN-1K | Sup. | Hie. | 224×224 | 88.7M | 15.4G | 645 | 84.3 |
| ■ Swin-S | IN-1K | Sup. | Hie. | 224×224 | 50M | 8.7G | 934 | 83.0 |
| ■ Swin-B | IN-1K | Sup. | Hie. | 224×224 | 88M | 15.4G | 710 | 83.5 |
| ■ DeiT-Ti | IN-1K | Sup. | Iso. | 224×224 | 5.7M | 1.3G | 6051 | 72.2 |
| ■ DeiT-S | IN-1K | Sup. | Iso. | 224×224 | 22.1M | 4.6G | 3080 | 79.8 |
| ■ DeiT-B | IN-1K | Sup. | Iso. | 224×224 | 86.4M | 17.6G | 1348 | 81.8 |
| ■ ViT-B/16 | IN-21K, IN-1K | Sup., Sup. | Iso. | 384×384 | 86.4M | 55.5G | 349 | 84.0 |
| ■ ViT-L/16 | IN-21K, IN-1K | Sup., Sup. | Iso. | 384×384 | 304.1M | 191.2G | 124 | 85.2 |
| ♣ PoolFormer-S12 | IN-1K | Sup. | Hie. | 224×224 | 12M | 1.8G | 4354 | 77.2 |
| ♣ PoolFormer-M48 | IN-1K | Sup. | Hie. | 224×224 | 73M | 11.6G | 768 | 82.5 |
| ♣ VanillaNet-5 | IN-1K | Sup. | Hie. | 224×224 | 15.5M | 5.2G | - | 72.5 |
| ♣ VanillaNet-13-1.5× | IN-1K | Sup. | Hie. | 224×224 | 127.8M | 26.5G | - | 82.5 |
| ⌖ AIM-0.6B | DFN-2B+, IN-1K | AR., Sup. | Iso. | 224×224 | 0.6B | - | - | 78.5 |
| ⌖ AIM-3B | DFN-2B+, IN-1K | AR., Sup. | Iso. | 224×224 | 3B | - | - | 82.2 |
| ⌖ ViM-B | IN-1K, IN-1K | Sup., Sup. | Iso. | 224×224 | 98M | - | - | 83.2 |
| ⌖ ViL-B | IN-1K, IN-1K | Sup., Sup. | Iso. | 224×224 | 89M | 18.6G | - | 82.4 |
| ⌖ VMamba-B | IN-1K | Sup. | Hie. | 224×224 | 89M | 15.4G | - | 83.9 |
| ⌖ P-VisionLLaMA-S | IN-1K | Sup. | Hie. | 224×224 | 24M | - | - | 81.6 |
| ⌖ P-VisionLLaMA-L | IN-1K | Sup. | Hie. | 224×224 | 99M | - | - | 83.6 |
| ⌖ VisionLLaMA-L | IN-1K, IN-1K | Sup., Sup. | Iso. | 224×224 | 310M | - | - | 84.6 |
| ★ iLLaMA-T | IN-1K | Sup. | Iso. | 224×224 | 5.7M | 1.3G | 6958 | 75.0 |
| ★ iLLaMA-S | IN-1K | Sup. | Iso. | 224×224 | 21.9M | 4.6G | 3222 | 79.9 |
| ★ iLLaMA-B | IN-1K | Sup. | Iso. | 224×224 | 86.3M | 17.6G | 1345 | 81.6 |
| ★ iLLaMA-B | IN-1K | Sup. | Iso. | 384×384 | 86.3M | 55.5G | 332 | 83.0 |
| ★ iLLaMA-B | IN-21K, IN-1K | Sup., Sup. | Iso. | 224×224 | 86.3M | 17.6G | 1345 | 83.6 |
| ★ iLLaMA-B | IN-21K, IN-1K | Sup., Sup. | Iso. | 384×384 | 86.3M | 55.5G | 332 | 85.0 |
| ★ iLLaMA-L | IN-21K, IN-1K | Sup., Sup. | Iso. | 224×224 | 310.2M | 62.8G | 456 | 84.8 |
| ★ iLLaMA-L | IN-21K, IN-1K | Sup., Sup. | Iso. | 384×384 | 310.2M | 194.7G | 116 | 86.0 |

size: After comprehensive pre-training on the expanded ImageNet-21K dataset, iLLaMA-B achieves more than $85.0\%$ accuracy on ImageNet-1K with under 100M parameters, significantly outperforming ViT-B's $84.0\%$. Upon scaling up to the larger iLLaMA-L, accuracy reaches $86.0\%$, exceeding that of ViT-L pre-trained on ImageNet-21K and the AIM-7B pre-trained on the DFN-2B+ dataset. To our knowledge, this showcases state-of-the-art performance for LLaMA-type architectures.

### 4.3 MODEL CALIBRATION AND SHAPE-TEXTURE BIAS

Beyond ImageNet accuracy, we also examined iLLaMA's calibration properties and shape-texture bias for a more detailed evaluation. Besides iLLaMA, we also explore two prevalent architectures, *i.e.*, ConvNeXt and DeiT3 (Touvron et al., 2022), representing ConvNets and Transformers, respectively. We apply ImageNet-21K pre-trained and ImageNet-1K fine-tuned models in this section.

**Model calibration.** Model calibration represents the relationship between a model's precision and confidence across samples of varying difficulty, *i.e.*, poor-calibrated models tend to produce overly confident yet incorrect predictions, whereas well-calibrated models demonstrate a strong correlation between confidence and accuracy (Guo et al., 2017). Calibration is commonly measured using the Expected Calibration Error (ECE), where a lower ECE is favorable. ECE results for different models on ImageNet-1K are presented in Table 8. The calibration of iLLaMA is lower than that of DeiT3, suggesting a more reliable output confidence. We also plot the reliability diagrams (Vishniakov et al., 2023) to intuitively compare the calibration of different models, as shown in Appendix G.

**Shape-texture bias.** Shape-texture bias measures the extent to which the model relies on the shape or texture of the image when performing recognition (Geirhos et al., 2018). We generally prefer

Table 7: **Quantization results.** #Bits: w bit weights, a bit activations. 8-bit iLLaMA-T matches 32-bit DeiT-T.

| Model | #Bits | Tiny | Small |
|---|---|---|---|
| DeiT | 32-32 | 72.2 | 79.8 |
| iLLaMA | 32-32 | 75.0 | 79.9 |
| iLLaMA | 8-8 | 72.4 | 77.4 |

Table 8: **Calibration** (expected calibration error ↓) and **shape-texture bias** (ratio ↑) results of ConvNeXt-B, DeiT3-B and iLLaMA-B. We use both IN-21K pre-trained and IN-1K fine-tuned models.

| Evaluation | ConvNeXt-B | DeiT3-B | iLLaMA-B |
|---|---|---|---|
| Calibration | 0.0281 | 0.0415 | 0.0335 |
| Shape-Texture Bias | 33.30% | 39.86% | 41.45% |

Table 9: **CIFAR transfer learning.** Soft mask improves iLLaMA performance without changing the inference architecture.

| Model | CIFAR10 | CIFAR100 |
|---|---|---|
| ViT-T | 98.0 | 85.5 |
| iLLaMA-T | 97.9 | 84.8 |
| + soft mask | 97.9 | 85.5 |

Table 10: **ADE20K semantic segmentation results using UperNet.** We report mIoU with multi-scale testing. FLOPs calculation are based on input sizes of (512, 512).

| Backbone | Input Crop. | mIoU | #Param. | FLOPs |
|---|---|---|---|---|
| ViT-T | $512^2$ | 39.8 | 10.88M | 37.1G |
| iLLaMA-T | $512^2$ | 37.7 | 10.86M | 37.1G |
| ViT-B | $512^2$ | 47.3 | 163.29M | 585.7G |
| iLLaMA-B | $512^2$ | 45.1 | 163.22M | 585.7G |

models to mimic human eye behavior, relying more on shape rather than texture (Tuli et al., 2021; Geirhos et al., 2020). We calculate the shape ratio for all models on cue-conflict images and report the results in Table 8, following (Vishniakov et al., 2023). Our iLLaMA shows the largest shape ratio of 41.45% among the three compared baselines, suggesting the potential of the LLM architecture for vision. Detailed results are provided in Appendix H.

## 4.4 COMPATIBILITY WITH QUANTIZATION

Since a practical goal for neural networks is deployment on low-bit hardware chips, we further examine iLLaMA's compatibility with quantization. We basically follow Q-ViT (Li et al., 2022) to apply quantization-aware training (QAT) to iLLaMA, with weights and activations of all blocks' FFN and causal self-attention layers to 8 bits. Quantization recipes and results are shown in Table 15 of Appendix C.4 and Table 7. Different sizes of low-bit iLLaMA maintain accuracy well, and 8-bit iLLaMA-T is even compete favorably with the full-precision DeiT-T (72.4% v.s. 72.2%).

## 4.5 TRANSFERABILITY ON DOWNSTREAM TASKS

**CIFAR transfer learning.** We fine-tune ViT-T and iLLaMA-T on the CIFAR datasets (Krizhevsky et al., 2009), including an ablation of the soft mask on iLLaMA. Detailed recipes are shown in Appendix C.5. iLLaMA's performance on CIFAR datasets is on par with ViT, assuring that iLLaMA can be confidently applied in the transfer learning field as a practical alternative to ViT. Additionally, soft mask is helpful in the relatively complicated CIFAR100, demonstrating its generalizability.

**ADE20K semantic segmentation.** We fine-tune our ImageNet-1K pre-trained iLLaMA and ViT models on ADE20K (Zhou et al., 2019) dataset using UperNet (Xiao et al., 2018) to perform semantic segmentation. For both iLLaMA and ViT, we set the learning rate as 6e-5 and weight decay as 0.01. Table 10 presents the results. iLLaMA's performance is marginally lower than ViT's, which we attribute to the potential impact of the masking mechanism in iLLaMA's causal attention on high-resolution dense prediction tasks. This suggests there is still space for optimization, a subject for future investigation.

## 5 CONCLUSIONS

In this paper, we systematically studies whether Transformer decoder, an architecture that has shown amazing potential in LLMs, can also take root in learning visual representation through straightforward supervised training. The key component – causal self-attention we used – is not novel and is inherited from existing LLM architectures, but we propose pivotal techniques, *i.e.*, PS [cls] and soft mask strategies, to effectively adapt them to visual tasks. The proposed iLLaMA outperforms many ConvNets, ViTs, and MLPs on imagenet, and demonstrates robust quantization compatibility, calibration, and shape-texture bias, thereby showing its practicality. We hope that this work will inspire a rethinking of generic yet practical architecture that can fully unify both vision and text.

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

## A NETWORK CONFIGURATION

In Table 11, we provide detailed architecture configurations for iLLaMA models of various capacities. Our approach to scaling up the model size, from small to large, is similar to that of the ViT. Thus, akin to ViT, iLLaMA benefits from the simplicity of an isotropic architecture and high throughput, with its internal features remaining unchanged in resolution and number of channel as the depth increases.

We provide a block-level comparison between iLLaMA and ViT model in Figure 5. VisionLLaMA uses SwiGLU, and AS2D RoPE to build LLaMA-style architecture. Differently, we further uses RMSNorm, modified causal self-attention and 1D RoPE from LLaMA to replace layer normalization, bi-directional self-attention, and proposes two pivotal strategies, *i.e.*, *PS [cls]* and *soft mask* to help the optimization of our iLLaMA. We also keep the learnable positional embedding as ViT.

Table 11: **iLLaMA architecture configurations.**

|  | Tiny (T) | Small (S) | Base (B) | Large (L) |
| --- | --- | --- | --- | --- |
| depth | 12 | 12 | 12 | 24 |
| embedding dim | 192 | 384 | 768 | 1024 |
| number of heads | 3 | 6 | 12 | 16 |
| #param. (M) | 5.7 | 21.9 | 86.3 | 310.2 |
| MACs (G) | 1.3 | 4.6 | 17.6 | 62.8 |

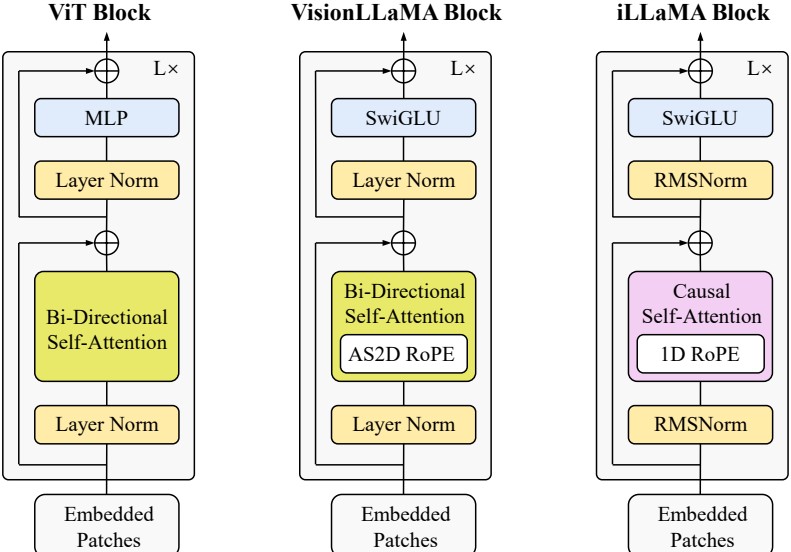

Figure 5: **Block level comparison.** We compare ViT (Dosovitskiy et al., 2020), VisionLLaMA (Chu et al., 2024), and iLLaMA blocks.

## B PYTORCH-LIKE CODE OF iLLaMA CAUSAL SELF-ATTENTION

The PyTorch-like implementation of our iLLaMA causal self-attention is shown as Algorithm 1. The iLLaMA code exhibits a high degree of similarity in structure and composition to the official LLaMA code [2] released by Meta, potentially offering considerable coding cost savings in developing a unified vision and language network with such architecture.

## C EXPERIMENTAL SETTINGS

### C.1 TRAINING RECIPE IN SECTION 3

Our training recipe for training the tiny and base models for Section 3 is primarily adapted from ConvNeXt (Liu et al., 2022) and early dropout (Liu et al., 2023), summarized in Table 12.

---

[2] https://github.com/meta-llama/llama

**Algorithm 1** PyTorch code of iLLaMA causal self-attention

```python
import torch
import torch.nn as nn

def reshape_for_broadcast(freqs_cis: torch.Tensor, x: torch.Tensor):
    ndim = x.ndim
    assert 0 <= 1 < ndim
    assert freqs_cis.shape == (x.shape[1], x.shape[-1])
    shape = [d if i == 1 or i == ndim - 1 else 1 for i, d in enumerate(x.shape)]
    return freqs_cis.view(*shape)

def apply_rotary_emb(
    xq: torch.Tensor,
    xk: torch.Tensor,
    freqs_cis: torch.Tensor,
) -> Tuple[torch.Tensor, torch.Tensor]:
    xq_ = torch.view_as_complex(xq.float().reshape(*xq.shape[:-1], -1, 2))
    xk_ = torch.view_as_complex(xk.float().reshape(*xk.shape[:-1], -1, 2))
    freqs_cis = reshape_for_broadcast(freqs_cis, xq_)
    xq_out = torch.view_as_real(xq_ * freqs_cis).flatten(3)
    xk_out = torch.view_as_real(xk_ * freqs_cis).flatten(3)
    return xq_out.type_as(xq), xk_out.type_as(xk)

class Attention(nn.Module):
    def __init__(self, dim, num_heads=8, qkv_bias=False, qk_scale=None, attn_drop=0.,
            proj_drop=0.):
        super().__init__()
        self.num_heads = num_heads
        head_dim = dim // num_heads
        # NOTE scale factor was wrong in my original version, can set manually to be
            compat with prev weights
        self.scale = qk_scale or head_dim ** -0.5

        self.qkv = nn.Linear(dim, dim * 3, bias=qkv_bias)
        self.proj = nn.Linear(dim, dim)

    def forward(self, x: torch.Tensor, freqs_cis: torch.Tensor, mask: Optional[torch.
            Tensor]):
        B, N, C = x.shape
        qkv = self.qkv(x).reshape(B, N, 3, self.num_heads, C // self.num_heads).permute(2,
            0, 1, 3, 4) # [3, B, N, self.num_heads, C // self.num_heads]
        q, k, v = qkv[0], qkv[1], qkv[2] # make torchscript happy (cannot use tensor as
            tuple) # [B, N, self.num_heads, C // self.num_heads]

        q, k = apply_rotary_emb(q, k, freqs_cis=freqs_cis)

        q = q.transpose(1, 2) # [B, self.num_heads, N, C // self.num_heads]
        k = k.transpose(1, 2) # [B, self.num_heads, N, C // self.num_heads]
        v = v.transpose(1, 2) # [B, self.num_heads, N, C // self.num_heads]
        attn = (q @ k.transpose(-2, -1)) * self.scale # [B, self.num_heads, N, N]
        attn = attn.softmax(dim=-1)
        if mask is not None:
            attn = attn * mask # (B, H, N, N)

        x = (attn @ v).transpose(1, 2).reshape(B, N, C)
        x = self.proj(x)

        return x
```

Basically, both regimes use the same experimental setup, with the only difference being the stochastic depth rate at 0.0 and 0.4, respectively. Notably, for the ViT baseline, our experimental results are 73.8% and 81.5%, as shown in Table 17, which slightly differ from the results of 73.9% and 81.6% reported in (Liu et al., 2023).

Utilizing only the basic training recipe with architectural modifications, the performance of iL-LaMA's tiny and base models achieves 73.2% and 81.2%, as shown in Table 17, yet remains below the ViT baseline. We attribute this to the impairing effect of causal self-attention on the information mixing among tokens. Thus, we enhance the training recipe, detailed next.

## C.2 IMAGENET (PRE-)TRAINING RECIPE

As illustrated in Table 13, we provide the detailed ImageNet-1K training hyper-parameters and ImageNet-21K pre-training hyper-parameters for the experimental results in Table 6.

Table 12: **Training settings.** We report details for Section 3 in the main paper, adapted from (Liu et al., 2023).

| Training Configuration | iLLaMA-T/B |
|---|---|
| *Initialization:* | |
| weight init | trunc. normal (0.2) |
| *Training recipe:* | |
| optimizer | AdamW (Loshchilov & Hutter, 2019) |
| optimizer momentum | $\beta_1, \beta_2 = 0.9, 0.999$ |
| *Learning hyper-parameters:* | |
| base learning rate | 4e-3 |
| learning rate schedule | cosine decay |
| weight decay | 0.05 |
| batch size | 4096 |
| training epochs | 300 |
| lr warmup epochs | 50 |
| warmup schedule | linear |
| gradient clip | None |
| exp. mov. avg. (EMA) (Polyak & Juditsky, 1992) | None |
| *Dropout:* | |
| dropout rate (Hinton et al., 2012) | 0.0 |
| stochastic depth rate (Huang et al., 2016) | 0.0/0.4 |
| *Data augmentation:* | |
| input resolution | $224^2$ |
| randAugment (Cubuk et al., 2020) | (9, 0.5) |
| random erasing (Zhong et al., 2020) | 0.25 |
| label smoothing (Szegedy et al., 2016) | 0.1 |
| mixup (Zhang et al., 2018) | 0.8 |
| cutmix (Yun et al., 2019) | 1.0 |

For the iLLaMA-T/S/B models, we train directly on ImageNet-1K and discover that models of different sizes are suited to different soft mask settings. For instance, the soft mask schedules are set to constant/linear/linear, respectively, with cutoff epochs designated as 50/50/25. We train the iLLaMA-T/S/B models using 8 A100 GPUs.

We pre-trained the iLLaMA-B/L models on ImageNet-21K for 90 epochs, adhering to the practices in (Liu et al., 2022). We set the cutoff epochs to 30, indicating that the iLLaMA models' self-attention fully transitions to causal self-attention after 30 epochs. We pre-train the iLLaMA-B/L models using 8 A100 GPUs.

### C.3    IMAGENET FINE-TUNING RECIPE

We present the results of fine-tuning models pre-trained on ImageNet-1K at a resolution of $384 \times 384$, as well as the outcomes of fine-tuning models pre-trained on ImageNet-21K at resolutions of $224 \times 224$ and $384 \times 384$, as shown in Table 14. All ImageNet-1K fine-tuning experiments were conducted for 30 epochs, following the convention in (Liu et al., 2022).

For the iLLaMA-B model pre-trained on ImageNet-1K, we used a relatively higher stochastic depth rate of 0.8. For the iLLaMA-B/L models pre-trained on ImageNet-21K, we employed relatively lower stochastic depth rates of 0.2 and 0.3, respectively.

Additionally, we standardized the cutoff epoch at 0 for our ImageNet-1K fine-tuning experiments, ensuring the application of a causal mask in self-attention to align with the LLaMA architecture. We also opted not to use learning rate warmup. We fine-tune the models using 8 A100 GPUs.

Table 13: **(Pre-)training settings.** We report details for iLLaMa model on ImageNet-1K/ImageNet-21K, respectively, adapted from (Liu et al., 2023). Some key training techniques are  highlighted .

| (Pre-)Training Configuration | iLLaMA-T/S/B ImageNet-1K | iLLaMA-B/L ImageNet-21K |
|---|---|---|
| *Initialization:* | | |
| weight init | trunc. normal (0.2) | trunc. normal (0.2) |
| *Training recipe:* | | |
| optimizer | AdamW | AdamW |
| optimizer momentum | $\beta_1, \beta_2 = 0.9, 0.999$ | $\beta_1, \beta_2 = 0.9, 0.999$ |
| *Learning hyper-parameters:* | | |
| base learning rate | 4e-3 | 1e-3 |
| learning rate schedule | cosine decay | cosine decay |
| weight decay | 0.05 | 0.01 |
| batch size | 4096 | 4096 |
| training epochs | 300 | 90 |
| warmup schedule | linear | linear |
| gradient clip | None | None |
| exp. mov. avg. (EMA) | None | None |
| *Dropout:* | | |
| dropout rate | 0.0 | 0.0 |
| stochastic depth rate | 0.0/0.1/0.4 | 0.1 |
| *Data augmentation:* | | |
| input resolution | $224^2$ | $224^2$ |
| randAugment | (9, 0.5) | (9, 0.5) |
| random erasing | 0.25 | 0.25 |
| label smoothing | 0.1 | 0.1 |
| mixup | 0.1/0.5/0.95 | 0.8 |
| cutmix | 0.1/0.5/1.0 | 1.0 |
| *Soft mask:* | | |
| soft mask schedule | constant/linear/linear | constant |
| cutoff epochs | 50/50/25 | 30 |
| lr warmup epochs | 5/5/50 | 5 |

### C.4 QUANTIZATION-AWARE TRAINING RECIPE

We provide our quantization-aware training recipe for iLLaMA in Table 15. Basically we follow the Q-ViT method proposed in (Li et al., 2022), with only weights and activations in each basic block's causal self-attention and FFN module are quantized to 8 bit width.

### C.5 CIFAR TRANSFER LEARNING RECIPE

We further provide our training recipe for transfer learning on the CIFAR10 and CIFAR100 datasets, as shown in Table 16. In our transfer learning experiments, we consistently apply a linear soft mask schedule. However, for the CIFAR10 and CIFAR100 datasets, we use cutoff epochs of 25 and 50.

## D DESIGNING iLLaMA: DETAILED RESULTS

We present the comprehensive experimental results of our exploration journey of iLLaMA in Table 17. This table not only delineates the stepwise accuracy of both the tiny and base models, as depicted in Figure 1, but also outlines the training loss at each step. The general trend observed is that as the training loss of the models decreases, their accuracy increases.

Overall, the trend in changes for the base model is broadly similar to that of the tiny model. However, in contrast to the tiny model, the implementation of RoPE coupled with subsequent integration of

Table 14: **Fine-tuning settings.** We report details for iLLaMa model on ImageNet-1K, adapted from (Liu et al., 2023). Some key training techniques are highlighted .

| (Pre-)Training Configuration | iLLaMA-B ImageNet-1K $224^2$ | iLLaMA-B/L ImageNet-21K $224^2$ | iLLaMA-B/L ImageNet-21K $224^2$ |
|---|---|---|---|
| Fine-Tuning Configuration | ImageNet-1K | ImageNet-1K | ImageNet-1K |
| *Initialization:* | | | |
| weight init | trunc. normal (0.2) | trunc. normal (0.2) | trunc. normal (0.2) |
| *Training recipe:* | | | |
| optimizer | AdamW | AdamW | AdamW |
| optimizer momentum | $\beta_1, \beta_2=0.9, 0.999$ | $\beta_1, \beta_2=0.9, 0.999$ | $\beta_1, \beta_2=0.9, 0.999$ |
| *Learning hyper-parameters:* | | | |
| base learning rate | 8e-5 | 8e-5/6e-5 | 1.1e-4/3.5e-5 |
| learning rate schedule | cosine decay | cosine decay | cosine decay |
| weight decay | 1e-8 | 1e-8 | 1e-8 |
| batch size | 512 | 512 | 512 |
| training epochs | 30 | 30 | 30 |
| warmup schedule | linear | linear | linear |
| gradient clip | None | None | None |
| exp. mov. avg. (EMA) | None | None | None |
| *Dropout:* | | | |
| dropout rate | 0.0 | 0.0 | 0.0 |
| stochastic depth rate | 0.8 | 0.2/0.3 | 0.2/0.3 |
| *Data augmentation:* | | | |
| input resolution | $384^2$ | $224^2$ | $384^2$ |
| randAugment | (9, 0.5) | (9, 0.5) | (9, 0.5) |
| random erasing | 0.25 | 0.25 | 0.25 |
| label smoothing | 0.1 | 0.1 | 0.1 |
| mixup | 0 | 0 | 0 |
| cutmix | 0 | 0 | 0 |
| *Soft mask:* | | | |
| soft mask schedule | constant | constant | constant |
| cutoff epochs | 0 | 0 | 0 |
| lr warmup epochs | 0 | 0 | 0 |

LPE does not affect the base model's performance. This lack of impact, we theorize, stems from the base regime's reduced susceptibility to underfitting compared to the tiny regime, hence the addition of extra learnable parameters offers less benefit to its performance.

Notably, vanilla causal self-attention proves inadequate for model optimization—the attention collapse issue effectively addressed by implementing the *PS [cls]* technique. Additionally, the application of the *soft mask* strategy significantly contributes to the training efficacy of both model sizes.

# E    RANK ANALYSIS OF CAUSAL SELF-ATTENTION

**Detailed visualization results.**    We provide rank analysis results of all 3 heads in layer 1, 4, 8, 12 of ViT-T/16 and iLLaMA-T/16 in Figure 11. We make four observations: 1) Not each head in each layer of iLLaMA's self-attention shows stronger concavity, suggesting that not every attention matrix of iLLaMA has a higher rank than its ViT counterpart. 2) In most cases, particularly in the shallow layers, the distribution of singular values in iLLaMA appears more uniform than in ViT. 3) In certain attention maps (*e.g.*, layer 8, head 2, and layer 8, head 3), the rank of ViT's attention matrix is low, resulting in an skewed distribution of information. In contrast, such extreme cases were not observed in our iLLaMA. 4) The distribution of singular values in ViT varies significantly

Table 15: **Quantization-aware training settings.** We report details for iLLaMa model on ImageNet-1K, adapted from (Liu et al., 2023; Li et al., 2022). Some key training techniques are highlighted .

| (Pre-)Training Configuration | iLLaMA-T/S ImageNet-1K |
|---|---|
| *Initialization:* | |
| weight init | trunc. normal (0.2) |
| *Training recipe:* | |
| optimizer | AdamW |
| optimizer momentum | $\beta_1, \beta_2 = 0.9, 0.999$ |
| *Learning hyper-parameters:* | |
| base learning rate | 3e-3/4e-3 |
| learning rate schedule | cosine decay |
| weight decay | 0.05 |
| batch size | 4096 |
| training epochs | 300 |
| warmup schedule | linear |
| gradient clip | None |
| exp. mov. avg. (EMA) | None |
| *Dropout:* | |
| dropout rate | 0.0 |
| stochastic depth rate | 0.0/0.1 |
| *Data augmentation:* | |
| input resolution | $224^2$ |
| randAugment | (9, 0.5) |
| random erasing | 0.25 |
| label smoothing | 0.1 |
| mixup | 0.1/0.5 |
| cutmix | 0.1/0.5 |
| *Soft mask:* | |
| soft mask schedule | constant/linear |
| cutoff epochs | 50/50 |
| lr warmup epochs | 5/5 |

Table 16: **Transfer learning settings.** We report details for ViT-T and iLLaMa-T model on CIFAR10/100, respectively, adapted from (Xu et al., 2024). Some key training techniques are highlighted .

| Transfer Learning Configuration | CIFAR10 | CIFAR100 |
|---|---|---|
| *For both ViT-T and iLLaMA-T:* | | |
| base learning rate | 2e-3 | 2e-3 |
| batch size | 1024 | 1024 |
| training epochs | 300 | 300 |
| stochastic depth rate | 0.0 | 0.0 |
| lr warmup epochs | 50 | 50 |
| *For iLLaMA-T only:* | | |
| soft mask schedule | linear | linear |
| cutoff epochs | 25 | 50 |

across different layers and heads (e.g., layer 1, head 1; layer 4, head 1; layer 8, head 1; layer 8, head 2), whereas iLLaMA's distribution appears relatively more stable.

Table 17: **ImageNet-1K classification accuracy.** We gradually replace components in ViT-T/16 and ViT-B/16 with counterparts in LLaMA, better or worse than the ViT baseline results with our basic training recipe. Components from or modified from LLaMA are highlighted. P.E.: positional embedding, Bd.: bi-directional self-attention, Cs.: causal self-attention.

| Ablation | FFN | Norm | Attention | P.E. | Tiny | Train Loss | Base | Train Loss |
|---|---|---|---|---|---|---|---|---|
| ViT | MLP | LN | Bd. | LPE | 72.2 | - | 81.8 | - |
| results with our basic training recipe | | | | | | | | |
| ViT | MLP | LN | Bd. | LPE | 73.8 | 3.451 | 81.5 | 2.828 |
| + LLaMa FFN | SwiGLU | LN | Bd. | LPE | 74.3 | 3.407 | 82.0 | 2.724 |
| + LLaMa Norm | SwiGLU | RMS | Bd. | LPE | 74.5 | 3.406 | 81.7 | 2.721 |
| + LLaMa S.A. | SwiGLU | RMS | Cs. | LPE | 0.1 | Failed | 0.1 | Failed |
| + LLaMa S.A. | SwiGLU | RMS | Cs. + *PS [CLS]* | LPE | 71.9 | 3.599 | 80.6 | 2.869 |
| + LLaMa P.E. | SwiGLU | RMS | Cs. + *PS [CLS]* | RoPE | 72.6 | 3.618 | 81.2 | 2.861 |
| + LPE P.E. | SwiGLU | RMS | Cs. + *PS [CLS]* | RoPE + LPE | 73.2 | 3.531 | 81.2 | 2.839 |
| modify the training techniques | | | | | | | | |
| + data aug. | SwiGLU | RMS | Cs. + *PS [CLS]* | RoPE + LPE | 74.3 | 2.990 | 81.3 | 2.868 |
| + *soft mask* | SwiGLU | RMS | Cs. + *PS [CLS]* | RoPE + LPE | **75.0** | 2.955 | **81.6** | 2.828 |

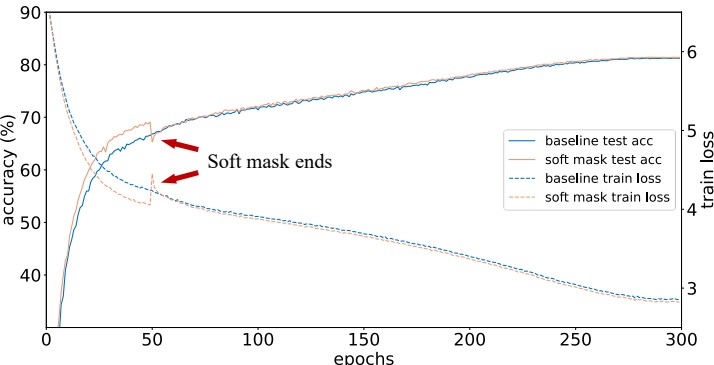

Figure 6: **Training curves.** Training curves for iLLaMA-B/16 regime w/ and w/o soft mask. When soft mask ends, the model experiences a similar pattern to the training curve of iLLaMA-T/16 regime, with eventually a lower test loss observed.

## F    ANALYSIS FOR SOFT MASK METHOD

In this section, we plot the training curves for iLLaMA-B/16 with and without the use of the soft mask strategy in Figure 6. We can observe that the results display a similar pattern to those of iLLaMA-T/16 (Figure 3(b)). We set the cutoff epochs to 50 and used a constant schedule. When soft mask ends, there is a sudden increase in training loss and a steep decline in model accuracy. However, the final accuracy surpasses the baseline, and the training loss is also optimized to a lower value. Such phenomenon shows the versatility of the soft mask across models of varying capacities, and shows that causal mode can achieve strong performance when a portion of attention is masked.

## G    MODEL CALIBRATION

To evaluate the calibration property, we plot the reliability diagrams of ConvNeXt-B, DeiT3-B and the proposed iLLaMA-B using ImageNet-1K in Figure 7, following (Vishniakov et al., 2023). For well-calibrated models, the direction of accuracy in their reliability diagrams show a roughly diagonal pattern, *i.e.*, the difference between accuracy and confidence is small. Intuitively, the confidence of the early bins of the iLLaMA presents results below the accuracy level, indicating that iLLaMA tends to be underconfident. This observation, similar to what was noted in DeiT3, may reflect a common feature of Transformer-based architectures, and was also noted in (Vishniakov et al., 2023).

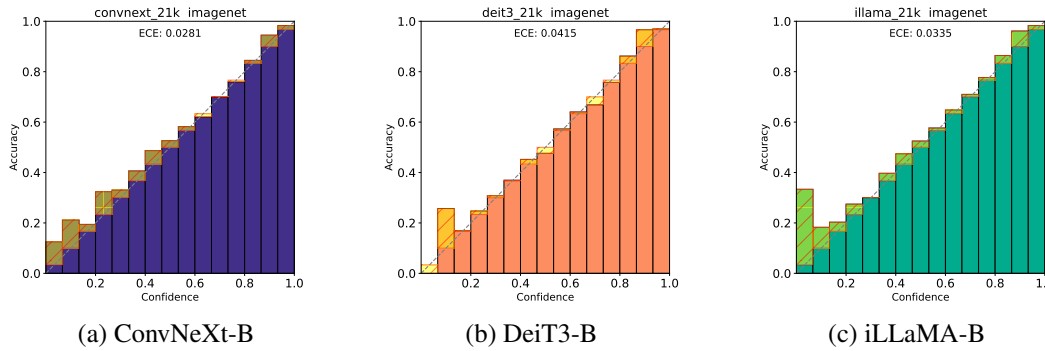

(a) ConvNeXt-B        (b) DeiT3-B        (c) iLLaMA-B

Figure 7: **Reliability diagram**. Calibration results of (a) ConvNeXt-B (b) DeiT3-B and (c) iLLaMA-B pretrained on ImageNet-21K and fine-tuned on ImageNet-1K.

## H  SHAPE-TEXTURE BIAS

We visualize the shape-texture bias results on cue-conflict images of ConvNeXt-B, DeiT3-B and the proposed iLLaMA-B in Figure 8, following (Vishniakov et al., 2023). The three dashed lines of different colors represent the average shape decision of the three models over all categories. Generally, a more leftward average shape ratio indicates that the model relies more on global shape information for recognition. iLLaMA shows higher shape scores compared to ConvNeXt and DeiT3.

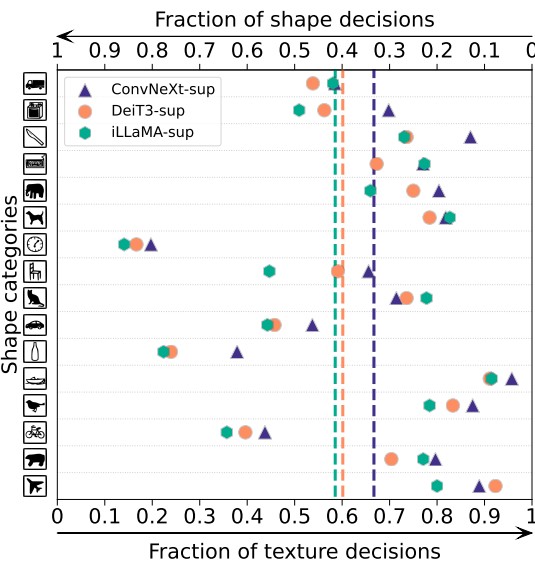

Figure 8: **Shape-texture bias.** Shape-texture bias results of ConvNeXt-B, DeiT3-B and iLLaMA-B pre-trained on ImageNet-21K and fine-tuned on ImageNet-1K. sup: supervised learning paradigm.

## I  INITIALIZATING iLLaMA USING PRE-TRAINED LLaMA

Previous studies (Zhang et al., 2024) have demonstrated that data unrelated to the image modality can be used to improve the performance of visual models. In fact, the pre-training dataset of LLaMA, which is entirely text, is irrelevant to the visual tasks that iLLaMA addresses. More important, the architectural components of iLLaMA are aligned with those of LLaMA. This alignment

Table 18: **Weight selection.** Results of iLLaMA initialization using LLaMA2-7B pre-trained weights.

| Model | Initialization | Tiny | Small | Base |
|-------|----------------|------|-------|------|
| iLLaMA | w/ weight selection | 74.5 | 79.9 | 81.4 |
| iLLaMA | w/o weight selection | 75.0 | 79.9 | 81.6 |

facilitates our exploration of using LLaMA's parameters to initialize iLLaMA, allowing us to fully exploit the potential of the weights of pre-trained LLMs.

We use the pre-trained LLaMA2-7B (Touvron et al., 2023b) to initialize our iLLaMA, instead of training from scratch. To match the dimensions of the weights, we employ the weight selection (Xu et al., 2024) method to initialize iLLaMA-T/S/B using a subset of the LLaMA2-7B pre-trained weights. Next, we train and evaluate the iLLaMA models, which are initialized using LLaMA2-7B, on the ImageNet-1K dataset. Other hyperparameter settings are consistent with Section 4.2. The results are shown in Table 18. Using LLaMA2 to initialize iLLaMA does not yield significant performance improvements. We attribute this to two main reasons: 1) The size difference between the two models is considerably large (LLaMA-2-7B's 7B parameters vs. iLLaMA-T's 5.7M parameters), resulting in a limited proportion of selected weights compared to meaningful pre-trained weights. 2) The training strategy was not adequately optimized. We believe that when using parameter inheritance, the corresponding training strategy should also be adjusted. However, we continued to use the training recipe designed for training from scratch.

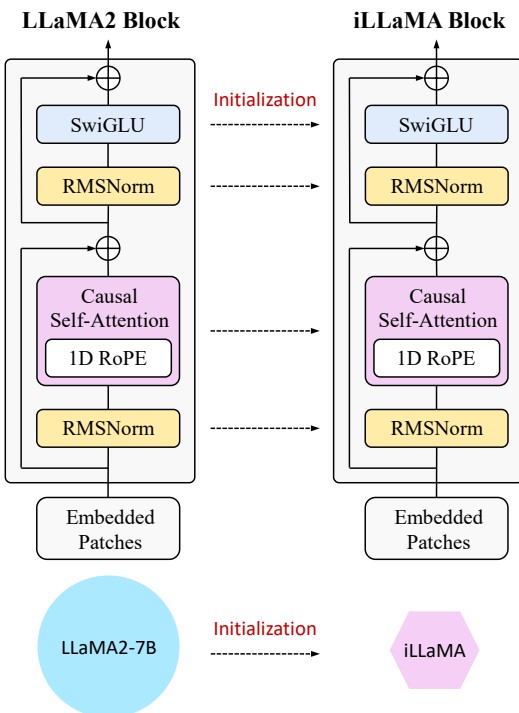

Figure 9: **Initializing iLLaMA using LLaMA.** iLLaMA initialization by pre-trained LLaMA2-7B (Touvron et al., 2023b) using weight selection (Xu et al., 2024).

## J  LOSS LANDSCAPE

As shown in Figure 10, we visualized the loss landscape (Li et al., 2018) of the iLLaMA-T/16 and ViT-T/16. The loss landscape of ViT and iLLaMA exhibits similar patterns, with the steepness and bumps observed in ViT seeming to be softened.

## K  CLASS ACTIVATION MAPS

In this section, we plot and compare the class activation map of representative models of several types of visual architectures, including ResNet-50 (He et al., 2016; Wightman et al., 2021), DeiT (Touvron et al., 2021), ConvNeXt (Liu et al., 2022), and iLLaMA, using GradCAM (Selvaraju et al., 2017). The results are shown in Figure 12. We find that iLLaMA's CAMs shows similar pattern to DeiT. We believe this stems from the attention-based architecture they share. We also

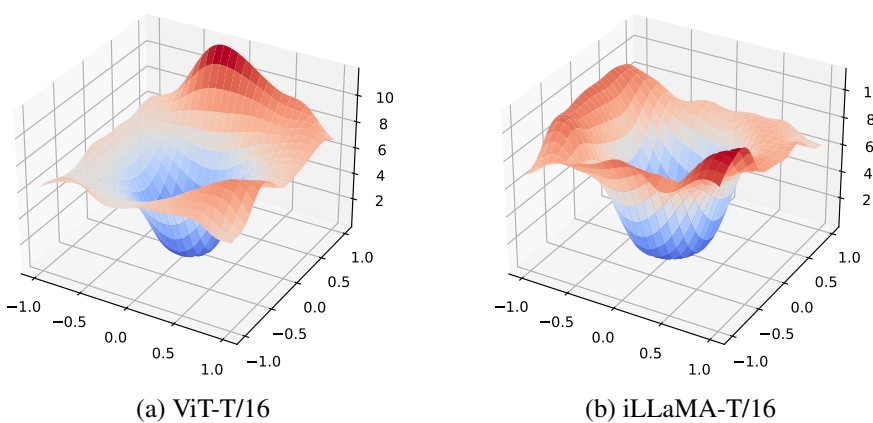

(a) ViT-T/16               (b) iLLaMA-T/16

Figure 10: **Loss landscape.** Illustration of (a) ViT-T/16 and (b) iLLaMA-T/16.

observe differences in the finer details between the CAMs of iLLaMA and DeiT, which can be attributed to the distinctions between causal mode attention and bi-directional one. We would like to note that the mechanism by which visual models perform classification remains a black box. It is not entirely clear which specific regions the model should focus on to achieve the correct results. Thus, we believe it is reasonable for iLLaMA to exhibit some unique patterns that differ from others.

## L    LIMITATIONS

We have shown that the LLaMA architecture, enhanced by the developed post-sequence [cls] and soft mask techniques, is adept at adapting to tasks such as visual recognition and semantic segmentation. However, iLLaMA's application remains predominantly within the realm of perception. In fact, such decoder-only architecture, favored by LLMs in the NLP field, can do more complex tasks, such as reasoning and generation. This may be due to their massive training data and the next sentence prediction training paradigm, which is not explored by iLLaMA. Thus, a critical validation step of aligning the architectures of text and visual models would be to construct a multi-modal large language model that fully leverages LLaMA components. In this envisioned model, both visual and textual feature extractors would be realized through the LLaMA architecture. Futhermore, we strongly argue that iLLaMA's successful attempts at basic supervised training strategies and classification tasks provide a foundation for more complex tasks, such as object detection and depth estimation. This represents a compelling avenue for future research.

## M    SOCIETAL IMPACT

After the ChatGPT milestone in 2022, open-source architectures like LLaMA began to shine in the text domain. In the real world, images and text are the two main mediums of information and data types. For neural networks, having a unified architecture for language and vision models allows people to process these two distinct types of information using the same structure, which aids in the specialization of hardware implementation. This paper transfers the architecture widely used in language models to vision models, facilitating the achievement of this goal. The pretrained models and code provided in this paper can be used in a plug-and-play manner to serve this objective.

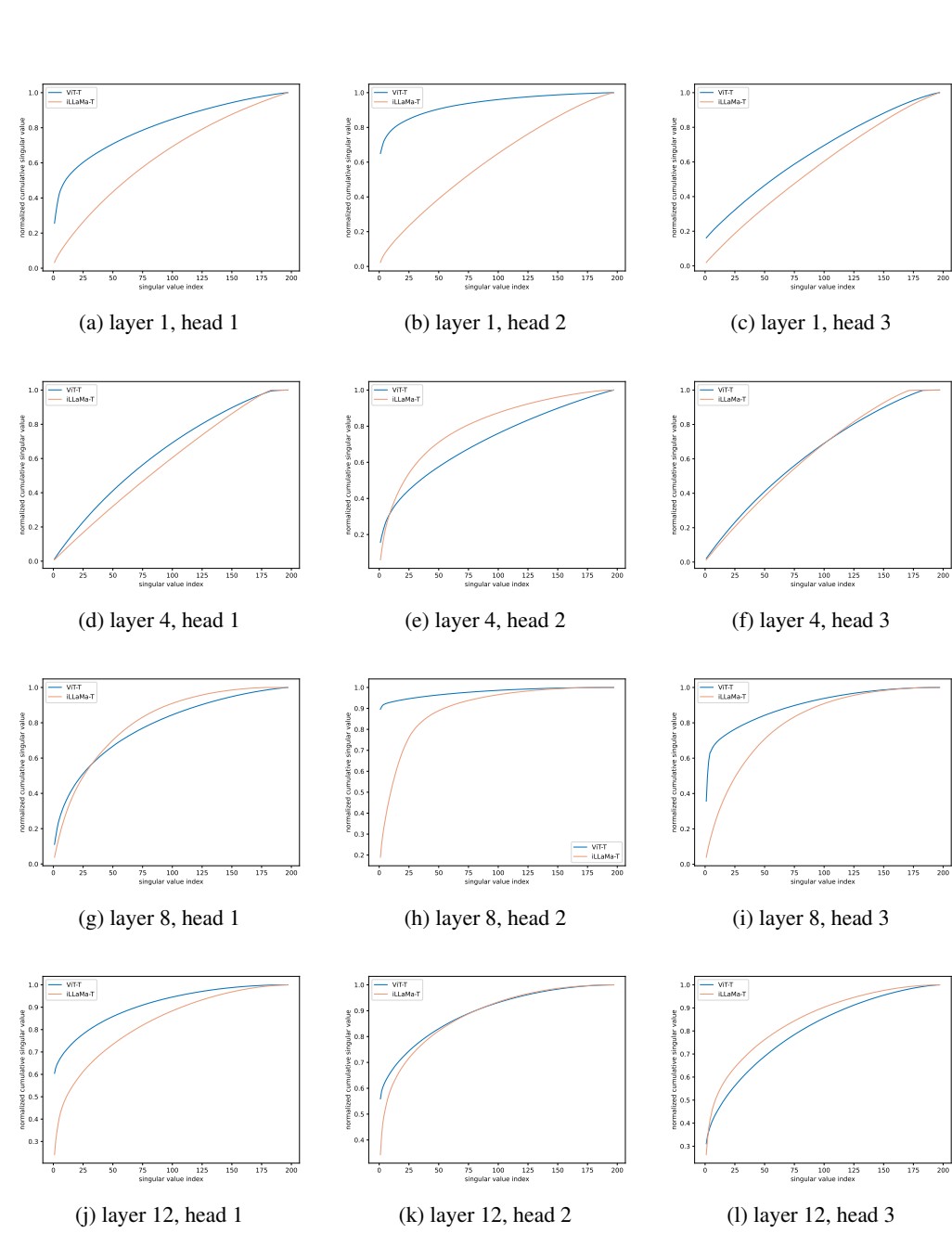

Figure 11: **Rank analysis.** Results of the self-attention matrix of all 3 heads in layer 1, 4, 8, 12 of the pretrained ViT-T and iLLaMA-T with $N = 197$. In most cases, especially in shallow layers, the singular values of iLLaMa show a more uniform distribution than ViT.

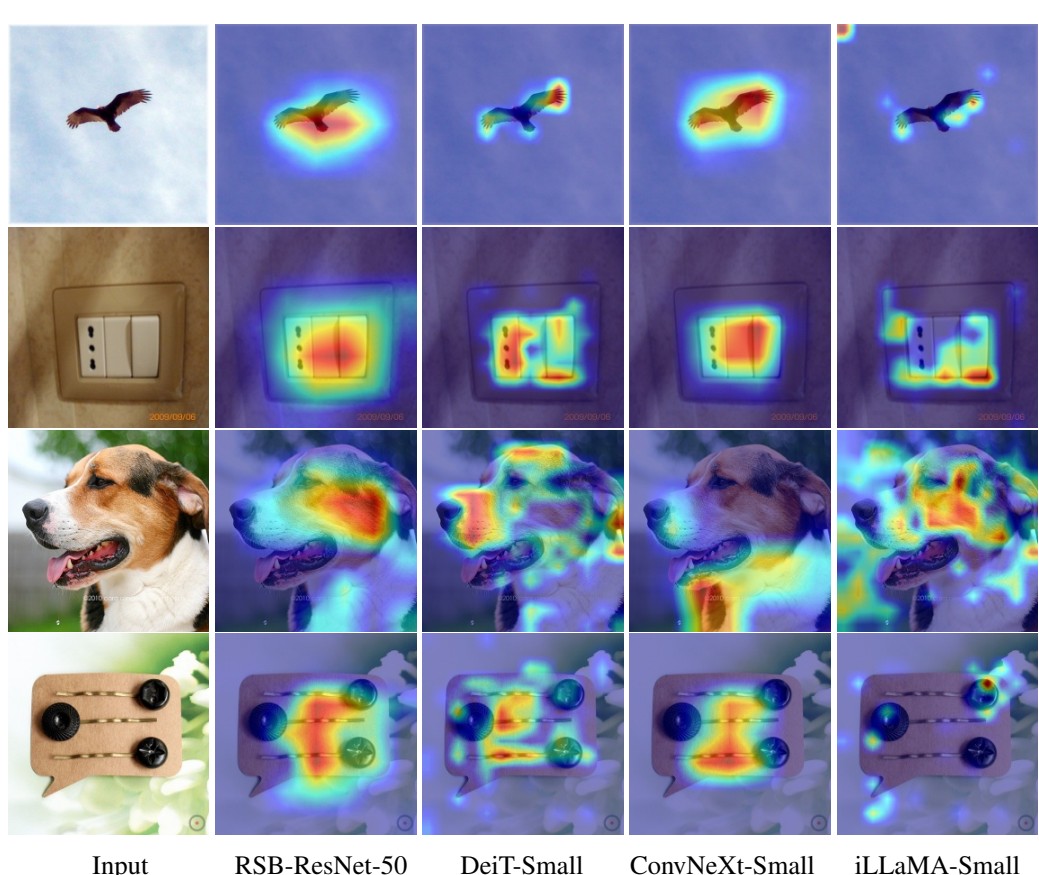

| Input | RSB-ResNet-50 | DeiT-Small | ConvNeXt-Small | iLLaMA-Small |

Figure 12: **Class activation maps.** Results are implemented by Grad-CAM of pre-trained models on ImageNet-1K dataset. The backbones include ResNet-50, DeiT-S, ConvNeXt-S, and our iLLaMA-S. Input images are sampled from the validation set.

