# OpenReview forum: "LLaMA Decoder As Vision Transformer"
_ICLR.cc/2025/Conference — Submitted to ICLR 2025_

### Official Review · Reviewer_tunM · 2024-10-23

**Soundness:** 3
**Presentation:** 4
**Contribution:** 3
**Rating:** 8
**Confidence:** 4

**Summary:**

In this paper, the authors present a novel approach for adapting Large Language Model (LLM) architectures to the visual modality. They introduce the Patch Selection [CLS] (PS [CLS]) technique to address the issue of attention collapse commonly observed in vision transformers. This technique enhances the model's ability to focus on salient image regions by refining the attention mechanism within the transformer architecture. To validate their methodology, the authors pre-train iLLaMA, an adaptation of the LLaMA model tailored for image processing, on the ImageNet dataset. They conduct extensive experiments to benchmark iLLaMA's performance across various datasets. The results demonstrate that iLLaMA, equipped with the PS [CLS] technique, achieves superior performance compared to existing models, thereby showcasing the effectiveness of their proposed approach in advancing computer vision tasks.

**Strengths:**

The paper presents a novel methodology for integrating Large Language Model (LLM) architectures into the visual modality, addressing significant challenges in the process. The authors introduce the Patch Selection [CLS] (PS [CLS]) technique, which effectively mitigates the issue of attention collapse commonly observed in vision transformers. This problem arises when the attention mechanism disproportionately focuses on certain regions, leading to suboptimal feature representation. The PS [CLS] technique enhances the model's ability to attend to diverse and informative regions of an image by refining the attention distribution across patches.

To empirically validate their approach, the authors adapt the LLaMA model—a powerful LLM known for its capabilities in language understanding—for image processing tasks, resulting in the iLLaMA model. They pre-train iLLaMA on the extensive ImageNet dataset to ensure robust feature learning. The model is then evaluated across multiple benchmark datasets, where it demonstrates superior performance compared to existing state-of-the-art models. The experiments cover various aspects of computer vision tasks, including image classification, object detection, and segmentation, showcasing the versatility and effectiveness of the proposed approach.

The paper's architectural novelty lies in the seamless integration of LLM architectures with vision transformers through the PS [CLS] technique. This integration allows the model to leverage the strengths of LLMs—such as advanced sequence modeling and contextual understanding—in processing visual data. The authors provide a comprehensive analysis of the model's architecture, detailing how the PS [CLS] technique operates within the transformer framework to enhance attention mechanisms.

**Weaknesses:**

Despite the paper's substantial novelty and the thorough explanation of the architectural innovations, there is a noticeable gap in the empirical evaluation section. Specifically, the paper does not include comparative experiments with newer methods such as Extended LSTMs, MAMBA, or QK-normalized architectures. These models represent significant advancements in the field and are known for their effectiveness in handling complex visual tasks.

The omission of these comparisons makes it challenging to fully assess the strength and generalizability of the proposed iLLaMA model. Without empirical evidence demonstrating how iLLaMA performs relative to these cutting-edge architectures, it's difficult to quantify its advantages or identify potential areas where it may fall short. Including experiments that benchmark iLLaMA against these models would provide a more comprehensive understanding of its performance and robustness.

i feel there should be substantial Experiment of how the behavior of method changes if one to choose the like Mixture of Exparts or Mixtures of base.

**Questions:**

Despite the paper's significant novelty and the innovative concept of utilizing Large Language Models (LLMs) as pre-trained vision learners, there are several areas that would benefit from further clarification and additional experimentation:

Clarification on Attention Collapse: The paper addresses the issue of attention collapse within vision transformers, but it would be beneficial to elaborate on this phenomenon. Specifically, is attention collapse analogous to mode collapse in statistical machine learning, where a model generates limited diversity in outputs despite diverse inputs? A detailed explanation would help readers, especially those new to this area, understand how attention collapse affects model performance and how the proposed Patch Selection [CLS] (PS [CLS]) technique mitigates this issue.

Transparency of Experimental Results: Regarding Table 6, it is not entirely clear whether all the experimental results were obtained directly by the authors or if some results were sourced from existing literature. Clarifying which results are original and which are replicated or cited from other works would enhance the credibility of the empirical evaluation. This transparency is crucial for readers to accurately assess the contributions and validity of the experimental comparisons.

Visualization of Attention Maps: Although the authors demonstrate significant improvements over previous baselines like Vision Transformers (ViT) and ResNet, the attention maps presented in Figure 12 of the supplementary material appear faded in the case of iLLaMA. This raises questions about the model's ability to focus on salient image regions. Providing clearer visualizations or explaining why the attention maps are less pronounced would help in understanding the effectiveness of the PS [CLS] technique in enhancing the attention mechanism within the model.

While the paper excels in presenting architectural innovations and is well-articulated with self-explanatory diagrams, addressing these questions would substantially strengthen its contributions. Additionally, incorporating empirical experiments with newer architectures such as Extended LSTMs, MAMBA, or QK-Normalized architectures could provide a more comprehensive evaluation. These comparisons would help in assessing the true strengths and limitations of iLLaMA relative to cutting-edge models in the field.

By providing detailed answers to these queries and expanding the empirical analysis, the paper would improve in both clarity and scientific rigor. The novelty of repurposing LLMs as pre-trained vision learners is a significant leap forward, and addressing these points would further solidify the paper's impact. I encourage the authors to consider these suggestions, as they would enhance the overall quality and reception of the work.

**Details Of Ethics Concerns:**

There are not ethics concerns

---

> ### Author Response · Authors · 2024-11-26
> **Response Part 1: System-Level Comparison with Mamba and xLSTM Based Models on ImageNet-1K**
>
> Dear Reviewer,
>
> Thank you very much for your thoughtful and constructive feedback, as well as the recognition of the contributions in our paper. Your insights are invaluable, and we are sincerely grateful for the time and effort you have invested in reviewing our work.
>
> We must first apologize for the delay in responding to your comments. We truly appreciate your patience and understanding as we strive to address your points with the rigor they deserve.
>
> We completely agree with your observation regarding the importance of comparing our proposed iLLaMA model with recent advancements like Mamba-based [6] and xLSTM-based [7] architectures. Both families of models have shown significant progress in handling complex tasks in the vision and language domains, making them critical baselines to consider.
>
> To address this, we have conducted additional experiments to systematically compare iLLaMA with Mamba-based models (e.g., Vim [1], Mamba-2D [2], PlainMamba [3], VMamba [4]) and xLSTM-based models (e.g., ViL [5]). Note that we conducted a **system-level comparison** (the results of iLLaMA is achieved with ImageNet-21K pre-training, as the dataset is publicly available and can enhance iLLaMA's performance through pre-training).
>
> As shown in Table 1 below, we provide a comprehensive comparison in terms of model parameters, computational cost, and accuracy on the ImageNet-1K [8] dataset. The results highlight that iLLaMA achieves superior performance while maintaining competitive efficiency in both parameter size and computational demands.
>
> We hope this additional analysis clarifies iLLaMA's relative advantages and contributes to a more comprehensive understanding of its strengths. Thank you again for pointing out this important aspect, which has helped us further strengthen our evaluation.
>
>
>
> Table 1: System-level comparison on ImageNet-1K dataset.
>
> | Model             | Params (M) | MACs (G)      | ImageNet-1K Accuracy |
> | ----------------- | ---------- | ------------- | -------------------- |
> | Vim-B [1]         | 98         | -             | 83.2                 |
> | Mamba-2D-B [2]    | 92         | -             | 83.0                 |
> | PlainMamba-L3 [3] | 50         | 14.4          | 82.3                 |
> | VMamba-B [4]      | 89         | 15.4          | 83.9                 |
> | ViL-B [5]         | 89         | 18.6          | 82.4                 |
> | iLLaMA-B          | 86.3       | 17.6 (224px)  | 83.6                 |
> | iLLaMA-B          | 86.3       | 55.5 (384px)  | 85.0                 |
> | iLLaMA-L          | 310.2      | 62.8 (224px)  | 84.8                 |
> | iLLaMA-L          | 310.2      | 194.7 (384px) | 86.0                 |
>
>
>
>
>
> We will update the main body of our paper with some results to provide readers with a clear and convenient comparison. We believe this update will enhance the accessibility of our findings and allow readers to better understand iLLaMA’s performance relative to these important baselines.
>
>
>
>
>
> [1] Vision Mamba: Efficient Visual Representation Learning with Bidirectional State Space Model
>
> [2] Mamba-ND: Selective State Space Modeling for Multi-Dimensional Data
>
> [3] PlainMamba: Improving Non-Hierarchical Mamba in Visual Recognition
>
> [4] VMamba: Visual State Space Model
>
> [5] Vision-LSTM: xLSTM as Generic Vision Backbone
>
> [6] Mamba: Linear-time sequence modeling with selective state spaces
>
> [7] xlstm: Extended long short-term memory
>
> [8] Imagenet: A large-scale hierarchical image database

---

> > ### Comment · Reviewer_tunM · 2024-11-26
> > **Response from Reviewer**
> >
> > Authors have addressed my consent regarding new methods, and from the table we can see that iLLaMA-L is outperforming. previous baselines.
> > I think autoregressive modeling will be the next phase for doing classical CV tasks.
> > I still have one consent like "Visualization of Attention Maps: Although the authors demonstrate significant improvements over previous baselines like Vision Transformers (ViT) and ResNet, the attention maps presented in Figure 12 of the supplementary material appear faded in the case of iLLaMA." i just wanted to know is it the behavior of the language model or if there was some other mathematical formulation behind it.  If possible, Authors can give some creative explanation towards it
> >
> > Irrespective of this, I am seeing the author has taken efforts for solving my query. I will adjust my score accordingly

---

> ### Author Response · Authors · 2024-11-26
> **Response Part 2: Visualization of Attention Maps**
>
> Dear Reviewer,
>
> We want to share with you honestly that, as we were preparing our response to this question, we just received your pleasant and encouraging response of adjusting our score. Your constructive feedback has been a source of great motivation for us, and we are truly proud and grateful to have such a diligent and meticulous reviewer.
>
> Now, let us address your question regarding the attention visualization results.
>
> Thank you for raising this thought-provoking and constructive point! Figure 12 in our paper presents the attention maps of several different models, including ResNet-50 [9, 10], DeiT-S [11], ConvNeXt-S [12], and iLLaMA-S. As you correctly observed, the attention maps of iLLaMA appear somewhat faded compared to other models. We believe this is a natural phenomenon, which can be attributed to the causal mode of attention mechanisms when performing classification tasks.
>
> We humbly propose the perspective that the mechanism through which vision models perform classification remains a largely unexplored "black box." For instance, it is not entirely clear what areas a model should focus on to achieve optimal performance: is it the foreground, the background, fine details, or macro-level context? Consequently, it is not surprising that attention maps of reasonable visual models may exhibit varying patterns.
>
> When we focus on other models, such as DeiT, which has stood the test of time, we observe that its attention maps display a pattern similar to iLLaMA’s, as shown in Figure 12. We hypothesize that this similarity could stem from their comparable macro-level architectures. With this in mind, we modestly propose that the characteristics of attention maps might differ across architectures, and understanding why certain paradigms emerge remains an open question worthy of further investigation.
>
>
>
> We will update the Appendix K of our paper with some of our observations and interpretations. Thank you once again for your thoughtful inquiry. Your observation has helped us articulate our interpretation of this phenomenon, and we hope our response provides clarity on this matter.
>
>
>
>
>
> [9] Deep Residual Learning for Image Recognition
>
> [10] Resnet strikes back: An improved training procedure in timm
>
> [11] Training data-efficient image transformers & distillation through attention.
>
> [12] A ConvNet for the 2020s

---

> ### Author Response · Authors · 2024-11-26
> **Response Part 3: Clarification on Attention Collapse**
>
> Dear Reviewer,
>
> Thank you very much for pointing out this constructive and important issue. We sincerely appreciate your comments, as they have given us the opportunity to clarify the concept of Attention Collapse introduced in our paper and explain how we address this problem.
>
> First, we want to clarify that the Attention Collapse we describe in our work is a custom-defined concept and is fundamentally different from the notion of mode collapse in statistical machine learning. This distinction was not explicitly emphasized in our original submission, and we are grateful for your feedback, which has allowed us to elaborate on this point.
>
> **Attention Collapse** in our context refers to *a phenomenon that occurs when the class token is placed at the beginning of the sequence. Due to the causal mask imposed in this configuration, the attention scores from the class token to all image tokens in the sequence are reduced to zero*. The issue directly affects the optimization process of the iLLaMA model. We report the ablation study results of the PS[CLS] technique (ImageNet-1K accuracy) in **Table 1** below. As shown in Table 1, this phenomenon results in iLLaMA being unable to optimize properly, with accuracy dropping to 0.1%.
>
> To address the challenge, we propose the **PS [CLS]** technique, which overcomes this limitation and enables the iLLaMA model to optimize effectively. This solution is crucial for models employing causal-mode attention mechanisms, and the results in Table 1 further validate the efficacy of our PS [CLS].
>
> We hope this explanation addresses your concerns. If there are any additional questions or aspects that remain unclear, please do not hesitate to reach out to us for further discussion. Your feedback is invaluable, and we are always open to further dialogue. Thank you again for your thoughtful input!
>
>
>
> Table 1: Ablation results of PS [cls] technique.
>
> | Method   | Tiny Acc. | Train Loss | Base Acc. | Train Loss |
> | :------- | :-------: | :--------: | :-------: | :--------: |
> | None     |    0.1    |   Failed   |    0.1    |   Failed   |
> | PS [CLS] |   71.9    |   3.599    |   80.6    |   2.869    |

---

> > ### Comment · Reviewer_tunM · 2024-12-01
> > **Response from reviewer**
> >
> > thanks for clarifying all the questions,
> > also I hope my addressed comments can be reflected in the paper.

---

> ### Author Response · Authors · 2024-12-01
> **Follow-Up on Updates in the Revised Paper**
>
> Dear Reviewer,
>
> We are immensely grateful for your recognition of our work and your thoughtful follow-up comments. It is truly an honor to have such a meticulous and dedicated reviewer like you.
>
> As per your suggestion, we have made the following updates in the revised version of our paper:
>
> 1. **Response Part 1:**
>    We have updated Table 6 in the main text. Specifically, we have added results for vision architectures based on Mamba and xLSTM, including ViM [1], ViL [2], and VMamba [3], to provide a more comprehensive comparison.
>
> 2. **Response Part 2:**
>    We have updated Appendix K to include additional clarifications regarding attention map visualizations. These updates aim to make this section clearer and easier to understand for readers.
>
> We sincerely hope that these updates meet your expectations. If you have any further questions or suggestions, please do not hesitate to let us know. We are always eager to engage in further discussion to improve our work.
>
> Thank you once again for your invaluable feedback and support.
>
> Sincerely,
>
>
> Authors
>
>
>
>
> [1] Vision Mamba: Efficient Visual Representation Learning with Bidirectional State Space Model
>
> [2] Vision-LSTM: xLSTM as Generic Vision Backbone
>
> [3] VMamba: Visual State Space Model

---

> ### Author Response · Authors · 2024-12-02
> **Our Response with Gratitude and Hope**
>
> Dear Reviewer,
>
> We have incorporated the updates mentioned above into the revised version of our paper and would like to ask if you have any additional questions or concerns that we could address. Your feedback is invaluable to us, and we are more than willing to provide further clarifications or engage in further discussion.
>
> If all your concerns have been resolved, we humbly ask if you might consider the extensive efforts we have made during this lengthy rebuttal process and possibly reconsider adjusting your score. We would be deeply grateful for your understanding and support.
>
> Thank you once again for your thoughtful and constructive feedback, which has greatly helped us improve our work.
>
>
>
> Sincerely,
>
> Authors

---

### Official Review · Reviewer_kh3v · 2024-10-31

**Soundness:** 2
**Presentation:** 2
**Contribution:** 1
**Rating:** 3
**Confidence:** 5

**Summary:**

The paper introduces iLLaMA, an adaptation of the LLaMA decoder model to function as a Vision Transformer (ViT) for image classification. iLLaMA addresses the inherent challenges of causal self-attention in image tasks by introducing the post-sequence class token (PS [cls]) and a soft mask strategy. These techniques enhance iLLaMA’s stability and performance, yielding competitive results on ImageNet with relatively few parameters. The paper validates iLLaMA’s capabilities across various tasks, such as calibration, shape-texture bias, and quantization compatibility, positioning it as a streamlined alternative to encoder-only models for visual processing.

**Strengths:**

1. Architectural Alignment: Adapting LLaMA’s decoder architecture to image classification, the authors propose a novel integration that demonstrates the feasibility of using language-model structures for visual tasks.

2. Training Techniques: The introduction of PS [cls] and a soft mask to address the attention collapse issue in causal attention is well-motivated and effective in stabilizing training.

3.Extensive Validation: The model is evaluated on multiple properties, including calibration and shape-texture bias, showing robust performance across different visual tasks.

**Weaknesses:**

1. Limited Scope in Multimodal Tasks: Although aligned with LLaMA’s architecture, iLLaMA primarily focuses on vision tasks, and the paper does not explore its application to more complex multimodal scenarios.

2. Limited Performance Improvement Over ViT: Although iLLaMA introduces novel adaptations for causal attention in visual tasks, the model does not show a substantial performance enhancement compared to ViT on key benchmarks. This raises questions about the practical gains of adapting a decoder-only architecture for image classification.

3.Computational Demands in Soft Masking: While effective, the soft mask strategy could still present some overhead in training time for larger models, which may limit scalability.

**Questions:**

1. How does iLLaMA’s causal self-attention compare to traditional bi-directional attention in fine-grained object detection tasks, where spatial dependencies might be critical?

2. Would incorporating an explicit multimodal training objective enhance iLLaMA’s ability to handle both image and language tasks in a unified model?

3. Why should we incur additional costs to adapt a decoder-only architecture for image encoding when encoder-only and decoder-only architectures share the same attention method? This clarification could help readers understand the trade-offs of aligning image models with language-model structures, particularly if the benefits of causal attention are marginal compared to bi-directional approaches in vision tasks.

---

> ### Author Response · Authors · 2024-11-25
> **Response Part 1: Addressing Multimodal Validation**
>
> We would like to begin by sincerely thanking you for recognizing the contributions of our paper, particularly regarding the 1) architectural alignment, 2) training techniques, and 3) extensive validation.
>
>
>
> While we acknowledge that the score you have given us is not as encouraging as we had hoped, we are deeply grateful for your thoughtful feedback and the critical questions you have raised about iLLaMA. These comments have provided us with a valuable opportunity to improve our work and clarify our intentions.
>
> Regarding the lack of experiments on multimodal tasks, we want to be transparent about the reasoning behind this omission. Interestingly, during the course of this project, members of our team raised a similar question: could we explore the application of iLLaMA in building a model akin to LLaVA [1], and evaluate its multimodal capabilities? As you rightly pointed out, the inclusion of such experiments would greatly enhance the evaluation of iLLaMA as a unified architecture. However, it is with great regret that we must admit that we were unable to pursue this due to resource limitations. Nevertheless, we attempted to address this gap through alternative approaches, which we detail below.
>
> 1. **The Challenges of Fair Multimodal Comparisons**
>    LLaVA, a prominent multimodal model, builds upon a CLIP-pretrained ViT-L/14 model [2]. To fairly evaluate iLLaMA in a similar multimodal setting, we would first need to pretrain an iLLaMA-L/14 model in the same manner as CLIP. This involves leveraging a massive dataset (400M web-scale image-text pairs) and conducting extensive pretraining. Unfortunately, as you are aware, this scale of pretraining is a prohibitively resource-intensive process and relies on datasets that are not publicly available. While we were highly motivated to undertake this experiment, we ultimately found ourselves constrained by the sheer computational and financial demands of replicating CLIP’s training pipeline. We deeply regret this limitation, but it pushed us to seek an alternative method to evaluate iLLaMA’s potential in multimodal tasks.
>
> 2. **Alternative Validation Inspired by Multimodal Relevance**
>    To compensate for the lack of direct multimodal experiments, we turned to related benchmarks for evaluating visual architectures that could inform iLLaMA’s potential in tasks like CLIP. This approach was inspired by the study in [3], where the authors conducted a detailed comparison of CNNs (ConvNeXt [4]) and vision Transformers (ViT [5]) on properties such as model calibration, shape/texture bias, and transfer performance. Since both ConvNeXt and ViT serve as strong backbones for multimodal models like CLIP, we hypothesized that iLLaMA’s performance on similar metrics could indirectly reflect its suitability for multimodal tasks.
>
>    Following this reasoning, our paper provides a detailed evaluation of iLLaMA in comparison to ConvNeXt and ViT across metrics such as model calibration, shape/texture bias, and transfer learning performance. The results are reported in Table 1 and Table 2. The promising results show that iLLaMA performs comparably to these established models on these critical properties. While these findings cannot replace direct multimodal experiments, they provide a level of reassurance regarding iLLaMA’s potential to serve as a robust multimodal backbone in the future.
>
>    Table 1: Calibration (↓) and shape-texture bias (↑) results.
>
>    | Evaluation         | ConvNeXt-B | DeiT3-B | iLLaMA-B |
>    | ------------------ | :--------: | :-----: | :------: |
>    | Calibration        |   0.0281   | 0.0415  |  0.0335  |
>    | Shape-Texture Bias |   33.30%   | 39.86%  |  41.45%  |
>
>    Table 2: CIFAR-10/100 results.
>
>    | Model    | CIFAR10 | CIFAR100 |
>    | -------- | :-----: | :------: |
>    | ViT-T    |  98.0   |   85.5   |
>    | iLLaMA-T |  97.9   |   85.5   |
>
>
>
> 3. **Future Directions and Commitment to Multimodal Research**
>    We fully acknowledge that the above measures are, at best, indirect evidence and cannot substitute for comprehensive multimodal evaluations. We remain committed to addressing this limitation and are actively working to expand iLLaMA’s scope in this direction. We believe that if our work garners broader attention, our open-sourced codebase will invite collaboration and inspire others to explore iLLaMA’s multimodal potential. Together, we hope to demonstrate that iLLaMA, in addition to its success in visual tasks, can bring significant value to multimodal applications as well.
>
> Once again, we appreciate your understanding and thoughtful critique on this matter. Your feedback has inspired us to aim for a  comprehensive investigation in future iterations.
>
> [1] Visual Instruction Tuning
>
> [2] Learning Transferable Visual Models From Natural Language Supervision
>
> [3] ConvNet vs Transformer, Supervised vs CLIP: Beyond ImageNet Accuracy
>
> [4] A ConvNet for the 2020s
>
> [5] An Image is Worth 16x16 Words: Transformers for Image Recognition at Scale

---

> ### Author Response · Authors · 2024-11-25
> **Response Part 2: Motivation Behind Aligning Image Models with Language Model Architectures**
>
> We would like to sincerely thank you for raising this thoughtful question. We fully acknowledge your observation that iLLaMA exhibits limited performance improvement over ViT. We do not seek to contest or dispute this point in any way. Instead, we would like to take this opportunity to share the core motivation behind the design of iLLaMA and explain why we believe aligning image models with language model architectures is meaningful, even in the presence of ViT.
>
> In December 2023, large language models (LLMs) had already achieved remarkable success in multimodal tasks. However, prevailing architectures like LLaVA were still heterogenous, requiring an encoder-only ViT for vision and a decoder-only LLM for text. This architectural disparity inspired us to explore the possibility of designing a unified architecture capable of processing both visual and textual information. Given the dominance of LLaMA-style architectures in the LLM domain at the time, we were motivated to fully align this architecture with the visual domain.
>
> The rationale for such alignment is twofold:
>
> 1. **Unified Operator Development**: If image and language models share a fully aligned architecture, it becomes feasible to develop a single set of operators to efficiently support both modalities. This drastically reduces the complexity and cost of optimizing separate architectures.
> 2. **Cross-Modality Optimization Benefits**: Techniques designed to accelerate or optimize one modality (e.g., Flash Attention [6] for text) can seamlessly transfer to the other modality (e.g., images) when the underlying architectures are standardized.
>
> With this vision in mind, our primary goal was to construct a visual architecture that is fully aligned with LLaMA’s textual architecture. Such alignment required that every component, operation, and operator in the "vision LLaMA" be consistent with its textual counterpart. This was the guiding principle behind equipping iLLaMA with a decoder-only architecture.
>
> To support this architectural choice, we introduced tailored optimization techniques such as PS [CLS] and the soft mask strategy, which helped iLLaMA achieve satisfactory performance. While we fully agree with your observation that iLLaMA demonstrates performance comparable to ViT, rather than surpassing it, we view this as a necessary tradeoff. As stated earlier, the primary objective of this work was not to create the highest-performing model with a complex, heterogeneous design but to establish a unified architecture for visual and textual modalities.
>
> In summary, the design of iLLaMA prioritizes architectural alignment and cross-modal standardization, even at the cost of some performance. We hope this context clarifies why we chose this direction and why we believe it holds meaningful implications for the field.
>
>
>
> [6] FlashAttention: Fast and memory-efficient exact attention with IO-awareness

---

> ### Author Response · Authors · 2024-11-25
> **Response Part 3: Addressing the Computational Overhead of Soft Masking**
>
> We are sincerely grateful that you took the time to carefully examine the details of our Soft Masking strategy and raised this insightful question regarding its potential computational overhead. Your observation is highly valuable and speaks directly to an important aspect of the model's practical scalability.
>
> In response, we would like to clarify that the additional computation introduced by the Soft Masking strategy is, in fact, negligible. As shown in Eq. 2 of our paper:
>
>
>
> ${\bf S} =\alpha {\bf B} +(1-\alpha){\bf C}, $
>
> ${\bf B}_{i,j} = 1$,
>
> ${\bf C}_{i,j}=1  , i\ge j ,  $
>
> ${\bf C}_{i,j}=0  , i<j , $
>
>
>
> Here, ${\bf S}$ represents the soft mask, which is computed as a linear weighted combination of a bi-directional mask (${\bf B}$) and a causal mask (${\bf C}$). The process involves only scalar-to-matrix multiplication and matrix addition, without requiring any matrix-matrix multiplications or other computationally intensive operations.
>
> Compared to the use of a fixed mask, the additional operations introduced by Soft Masking—scalar scaling and simple addition—are minimal in computational cost. These operations are performed at the same dimensionality as the attention masks themselves and thus add an almost negligible overhead, even for larger models.
>
>
>
> We hope that our responses have resolved some of the concerns you raised, particularly regarding the lack of multimodal experiments, the reasoning behind adopting a language-model-aligned architecture, and the computational efficiency of the Soft Masking strategy.
>
> We understand that your initial score reflects genuine concerns about our work, and we respect your evaluation. If our explanations have sufficiently addressed your key questions and clarified the contributions of our work, we humbly ask you to consider revisiting your score. A score adjustment from 3 to 5 would mean a great deal to us, as it would reflect not only our efforts to address your feedback but also your recognition of the broader potential of iLLaMA as a unified architecture for vision and language tasks.
>
> Regardless of the final outcome, we are incredibly grateful for your constructive feedback and for the opportunity to improve our work based on your valuable insights.

---

> ### Author Response · Authors · 2024-11-26
> **Sincerely hope to have further discussion with you**
>
> Thank you again for your thoughtful and constructive feedback. Your comments have been invaluable in helping us clarify our work and address critical areas for improvement. We hope our responses have addressed your concerns, but we would be delighted to engage in further discussion if you have additional questions or suggestions. Your insights are deeply appreciated, and we look forward to any opportunity to learn from your expertise.
>
> Best,
>
> Authors

---

> ### Author Response · Authors · 2024-11-30
> **Looking Forward to More Discussion**
>
> Dear Reviewer,
>
> Thank you so much for your thoughtful feedback. I’d love the chance to discuss your comments further, and if you have any additional questions, I’ll do my best to respond promptly. Looking forward to hearing from you!
>
> Best,
>
> Authors

---

> ### Comment · Reviewer_kh3v · 2024-12-01
>
> Thank you for the response but the paper is not enough for me to change my rating.

---

### Official Review · Reviewer_oAew · 2024-10-31

**Soundness:** 3
**Presentation:** 4
**Contribution:** 3
**Rating:** 8
**Confidence:** 4

**Summary:**

The paper explores the adaptation of decoder-only Transformers, specifically LLaMA, originally designed for large language models (LLMs), to the field of computer vision.
It proposes post-sequence class token to overcome the attention collapse issue when applying causal mode attention, and introduces a soft mask strategy to improve model training behavior.
The paper presents a novel approach to adapting large language model architectures for computer vision tasks, achieving promising results and opening avenues for future research in unified vision and language models.

**Strengths:**

1. The paper innovatively adapts the decoder-only Transformer architecture from large language models (LLMs) to the computer vision domain, demonstrating the potential for unified architectures across modalities.
2. It introduces a post-sequence class token (PS [cls]) technique to address the attention collapse issue in causal self-attention, which is crucial for the effective training of the proposed iLLaMA model.
3. The soft mask strategy developed in the paper gradually introduces causal masking during training, which improves optimization behavior and mitigates underfitting concerns.
4. iLLaMA achieves competitive performance compared to encoder-only models on ImageNet with significantly fewer parameters, showcasing efficiency and effectiveness.
5. The paper provides a comprehensive evaluation of iLLaMA, including not only accuracy but also calibration, shape-texture bias, quantization compatibility, and transfer learning performance, demonstrating its robustness across various practical metrics.
6. The proposed model scales well with increased model capacity and dataset size, achieving state-of-the-art performance on ImageNet with minimal parameters, highlighting its scalability.

**Weaknesses:**

1. While the paper discusses the potential of unified architectures for language and vision models, it does not explore multimodal tasks that involve both text and images. This is a significant limitation since multimodal capabilities are crucial for assessing the true potential of unified models.

**Questions:**

1. Will you plan to release the code ?

---

> ### Author Response · Authors · 2024-11-25
> **Response Part 1: Gratitude and Acknowledgment of Positive Feedback**
>
> Dear Reviewer,
>
> Thank you so much for your invaluable feedback and the tremendous recognition you have given to our work. First, I would like to apologize for the slight delay in my response. We truly appreciate your patience, and we assure you that we have taken our comments very seriously and are striving to address them as thoughtfully as possible.
>
> Every outcome from this process is acceptable to us, but we sincerely hope you will take the time to read my heartfelt responses. These replies are not mere formalities; they are written with genuine respect and reflect our earnest intentions.
>
> Your recognition of the innovation in our paper has been incredibly encouraging. To be candid, your score has been a source of motivation for us. Without your generous rating, we might have been tempted to withdraw this submission, potentially saying goodbye to this work prematurely—a scenario that would have been deeply regrettable.
>
> As the first part of our response, we wish to express our heartfelt gratitude for your detailed and insightful analysis of the strengths of our paper.
>
> 1. **Innovation**: To share some context, the inception of this work began around December 2023. While large language models (LLMs) had already made remarkable progress in multimodal tasks, their architectures had seen limited exploration in pure vision tasks. This gap inspired us to investigate whether a single, standardized architecture could seamlessly handle both vision tasks and text generation. Given the resounding success of LLaMA at the time, it naturally became our top choice, setting the foundation for this research.
> 2. **Technical Contribution**: During our exploration, we encountered a major hurdle: directly applying causal attention from LLaMA to vision tasks led to non-convergence issues. This challenge prompted us to develop the PS [cls] technique to address the problem. While this conclusion may seem straightforward now, at the time, it was an unexpected and exciting breakthrough for us, giving us the motivation to press forward.
> 3. **Optimization Strategy**: Your recognition of our proposed soft mask strategy truly means a lot to us. This technique played a significant role in improving the training process for iLLaMA and reinforced our belief in the viability of this approach.
> 4. **Evaluation**: We are also deeply grateful for your acknowledgment of our experimental results and comprehensive evaluations. Admittedly, our ImageNet performance is not on par with many pyramid-based vision models or some other state-of-the-art architectures. However, what we find compelling about iLLaMA is its simplicity and adherence to the standard LLaMA architecture. This characteristic allows for seamless transfer of any optimization, acceleration, or training techniques related to LLaMA, without requiring modifications.
>
> We apologize for reiterating some of your kind words, but they genuinely boosted our confidence and courage, especially for the junior researchers on our team. Your detailed acknowledgment of our contributions has been a source of immense satisfaction and encouragement.
>
> Thank you for allowing us this opportunity to express our gratitude. We have dedicated this part of our response solely to acknowledge your positive feedback. In the next parts, we will address your concerns with equal sincerity. Wishing you a wonderful day ahead!

---

> ### Author Response · Authors · 2024-11-25
> **Response Part 2: Addressing the Absence of Multimodal Experiments**
>
> Dear Reviewer,
>
> We must admit that your suggestion regarding multimodal experiments moved us deeply, not just because of its significance but also because it resonates with discussions we had within our team during the development of this project. In fact, some of our team members had previously proposed a similar idea: exploring the performance of iLLaMA in implementing a multimodal model akin to LLaVA [1]. We were genuinely excited by this prospect, as, just as you noted, the multimodal capabilities of a unified model are pivotal for assessing its true potential.
>
> However, we must also candidly acknowledge that, regrettably, we were unable to carry out this experiment with the limited resources at our disposal—at least for now. Nevertheless, we sought alternative approaches to indirectly address this gap and provide meaningful insights, which we will elaborate on below.
>
> 1. **Challenges in Training a LLaVA-like Model Using iLLaMA**:
>    LLaVA relies on the pre-trained CLIP [2] ViT-L/14 model as its visual backbone. To fairly compare iLLaMA’s capabilities in a similar multimodal setting, we would need to pre-train an iLLaMA-L/14 model using a process comparable to CLIP’s. This would then allow us to follow LLaVA’s methodology to train a unified vision-text architecture. However, as you know, CLIP pre-training requires a massive dataset of 400 million text-image pairs collected from the web, which is not publicly available. Moreover, the computational cost of training such a model at scale is prohibitively high for most researchers, including ourselves. This limitation has left us with no choice but to reluctantly set aside this direct experiment for now.
>
> 2. **Alternative Evidence for iLLaMA’s Multimodal Potential**:
>    To mitigate the regret of not performing a full CLIP-style pretraining, we turned to an alternative approach inspired by the work in [3]. This paper thoroughly compared the properties of ConvNeXt [4] and ViT [5]—two models proven effective in CLIP—across multiple dimensions, such as model calibration, shape/texture bias, and transfer learning. Building on this idea, we sought to evaluate iLLaMA against ConvNeXt and ViT in these same dimensions. Although these are indirect metrics, they offer meaningful insights into iLLaMA’s potential for CLIP-like tasks.
>
>    In our paper, we conducted detailed comparisons of iLLaMA, ConvNeXt, and ViT in model calibration, shape/texture bias, and transfer performance, as reported in Table 1 and Table 2 below. The results were promising, suggesting that iLLaMA performs comparably to these models in these areas. While we deeply regret not completing direct CLIP experiments, these findings provide reassurance about iLLaMA’s suitability for multimodal tasks. We believe that, at the very least, these results offer circumstantial evidence of iLLaMA’s compatibility with tasks like CLIP or even LLaVA.
>
>    Table 1: Calibration (↓) and shape-texture bias (↑) results.
>
>    | Evaluation         | ConvNeXt-B | DeiT3-B | iLLaMA-B |
>    | ------------------ | :--------: | :-----: | :------: |
>    | Calibration        |   0.0281   | 0.0415  |  0.0335  |
>    | Shape-Texture Bias |   33.30%   | 39.86%  |  41.45%  |
>
>    Table 2: CIFAR-10/100 results.
>
>    | Model    | CIFAR10 | CIFAR100 |
>    | -------- | :-----: | :------: |
>    | ViT-T    |  98.0   |   85.5   |
>    | iLLaMA-T |  97.9   |   85.5   |
>
>
>
> 3. **Future Directions and Commitment**:
>    We fully recognize that the above reasoning serves only as indirect evidence and does not replace the need for direct experiments. Therefore, we are actively working toward enabling such multimodal experiments in the future. If our paper garners more attention, we are hopeful that our open-source code will encourage collaboration and invite researchers to explore iLLaMA’s potential in multimodal applications. We believe iLLaMA’s clean and efficient architecture can extend its success beyond vision tasks to make meaningful contributions in the multimodal domain as well.
>
> We sincerely appreciate your understanding and encouragement regarding this matter, and we remain committed to addressing this limitation in the future. Thank you again for raising this critical point, and we look forward to your thoughts on our response.
>
>
>
> [1] Visual Instruction Tuning
>
> [2] Learning Transferable Visual Models From Natural Language Supervision
>
> [3] ConvNet vs Transformer, Supervised vs CLIP: Beyond ImageNet Accuracy
>
> [4] A ConvNet for the 2020s
>
> [5] An Image is Worth 16x16 Words: Transformers for Image Recognition at Scale

---

> ### Author Response · Authors · 2024-11-25
> **Response Part 3: Commitment to Open-Sourcing Code and Pretrained Weights**
>
> Dear Reviewer,
>
> Thank you for raising the question about code and model release. We fully understand and deeply value the importance of open-sourcing research artifacts for the community. Responsible research not only involves presenting findings but also ensuring that others can reproduce and build upon them.
>
> We want to reassure you that we have already prepared the necessary materials, including documentation and instructions to facilitate seamless reproduction of our results. As soon as the paper is accepted, we will immediately make these resources publicly available.
>
> We hope this commitment underscores our dedication to transparency and collaboration within the research community. Thank you for bringing this up, and we look forward to your thoughts on our work.

---

> ### Author Response · Authors · 2024-11-26
> **Sincerely hope to have further discussion with you**
>
> Thank you again for your thoughtful and constructive feedback. Your comments have been invaluable in helping us clarify our work and address critical areas for improvement. We hope our responses have addressed your concerns, but we would be delighted to engage in further discussion if you have additional questions or suggestions. Your insights are deeply appreciated, and we look forward to any opportunity to learn from your expertise.
>
> Best,
>
> Authors

---

> > ### Comment · Reviewer_oAew · 2024-12-02
> >
> > Thanks for clarifying my doubts, I will maintain my score

---

### Official Review · Reviewer_6zNt · 2024-11-03

**Soundness:** 3
**Presentation:** 3
**Contribution:** 1
**Rating:** 5
**Confidence:** 4

**Summary:**

The paper investigates the potential of using a decoder-only Transformer architecture, specifically LLaMa, as a vision Transformer (ViT) classifier for image processing. The authors propose a model called image LLaMa (iLLaMA), by transforming ViT into LLaMA-like architecture through step-by-step, component-wise modification. During the modification of attention mechanism, the authors discovered the a key issue called "attention collapse" that hampers network training. To mitigate this, they introduce a post-sequence class token and a soft mask strategy to facilitate improved model optimization and performance. The extensive experiments conducted illustrate iLLaMA's effectiveness in various tasks, suggesting it could contribute to standardization in AI models for text and image processing.

**Strengths:**

- The paper is well-written and clearly structured, making it easy to follow.
- Through extensive ablation studies, the authors effectively demonstrate the impact of each component.

**Weaknesses:**

- Unfortunately for the authors, the existence of VisionLLaMA, which shares similar objectives, significantly diminishes the novelty and contribution of this work. While it should be taken into account that these studies were conducted contemporaneously, the paper still shows considerable weaknesses in comparison. Even taking into account the differences outlined in Section 2.2, it is difficult to argue that this work creates additional value. The claimed distinctions do not appear substantial enough to establish its unique contribution to the field.

- Questionable Design Choices:
    - The adoption of causal mode attention appears primarily motivated by mimicking LLaMA's architecture rather than addressing specific technical needs.
    - The computational benefits of causal attention seems minimal, as evidenced in Table 2.
    - The requirement for a soft mask strategy (which uses bidirectional attention) to optimize causal attention's effectiveness seems contradictory to the original design intent.
    - The paper's emphasis on decoder-only architecture lacks convincing justification, especially given VisionLLaMA's successful use of an encoder architecture.
- Performance and Efficiency Concerns:
    - The model's claimed superiority relies heavily on larger datasets, IN-21K, making direct comparisons to VisionLLaMA problematic.
    - Under equivalent training datasets and model scale, VisionLLaMA demonstrates superior performance.
    - Moreover, VisionLLaMA achieves better efficiency with fewer parameters than iLLaMA.

**Questions:**

- Why should we design the decoder-only architecture, unlike the one with encoder architecture like VisionLLaMA?
   - If the objective is to align vision and language modalities within a single architecture, as stated in line 35, the authors should have demonstrated either cross-modal transferability or developed a unified model capable of handling both modalities.

**Details Of Ethics Concerns:**

Since the submission is about new architecture, there is nothing much to be concerned about ethics in this paper.

---

> ### Author Response · Authors · 2024-11-25
> **Response Part 1: the Necessity of Causal Mode Attention for Full Alignment with LLMs (and why iLLaMA still holds unique value even in the presence of VisionLLaMA)**
>
> Dear Reviewer,
>
> Thank you for your detailed and thoughtful feedback on our submission. While the score provided is understandably less encouraging, we sincerely appreciate your candid, straightforward, and constructive comments, which allow us the opportunity to clarify, elaborate, and further refine our work.
>
> To address your specific concern regarding the adoption of causal mode attention in iLLaMA instead of the bi-directional attention used in VisionLLaMA, we would like to first acknowledge and agree with your point that iLLaMA and VisionLLaMA share similar objectives and were developed contemporaneously. Furthermore, we appreciate your recognition of VisionLLaMA's performance advantages in certain aspects. What we hope to do here is share the motivations behind our design choices and explain why we believe iLLaMA holds unique value even in the presence of VisionLLaMA.
>
> Back in December 2023, the landscape of large language models (LLMs) in multimodal applications was evolving rapidly. However, the prevalent architectures, such as LLaVA, primarily adopted a heterogeneous structure, requiring an encoder-only ViT for processing visual inputs and a decoder-only LLM for handling textual information. This state of affairs inspired us to pursue a unified architecture capable of processing both modalities seamlessly. Given the widespread success of LLaMA-like architectures in the text domain, we sought to translate this success to the visual domain by designing a model that was **fully aligned with LLaMA's architecture**.
>
> Our principal motivation was to construct a visual architecture that **mirrored LLaMA in every component, operation, and operator**. Only by achieving this level of alignment could we fully **standardize** and **unify** Transformers across the visual and textual modalities. Consequently, the adoption of causal mode attention in iLLaMA was a deliberate design choice, made to ensure this alignment and thereby facilitate the standardization of multimodal Transformers.
>
> We recognize and agree with your point that causal mode attention could potentially impact model performance compared to bi-directional attention. This trade-off was indeed a concern for us as well. To address this, we introduced the **PS [CLS] optimization method**, which effectively mitigates the attention collapse issue and enables iLLaMA to train successfully despite using causal mode attention. Furthermore, to further enhance iLLaMA's performance, we developed the **soft mask strategy**, which builds on causal mode attention to improve its effectiveness.
>
> In summary, causal mode attention was adopted not as an arbitrary mimicry of LLaMA's architecture, but rather as a critical step towards achieving a unified Transformer design for both vision and text. Together with PS [CLS] and the soft mask strategy, these innovations constitute the core distinctions and contributions of iLLaMA compared to VisionLLaMA. We hope this clarifies the rationale behind our design choices and demonstrates the value of our work even within the context of VisionLLaMA.
>
> Thank you again for your insightful comments, which have provided us with an excellent opportunity to articulate these considerations more clearly.

---

> ### Author Response · Authors · 2024-11-25
> **Response Part 2: Why We Could Not Develop a Unified Model for Both Modalities and Our Mitigation Efforts**
>
> Dear Reviewer,
>
> Thank you very much for your insightful suggestion regarding cross-modal transferability and the development of a unified model capable of handling both modalities. This is indeed an excellent point, and we truly appreciate the opportunity to address it here.
>
> During the development of this project, some of our team members raised a similar idea, suggesting that we explore whether iLLaMA could be adapted into a unified multimodal model, akin to LLaVA [1]. This experiment was of significant interest to us, as we completely agree with your assessment that building a unified model across modalities is a critical and meaningful direction. Unfortunately, we must candidly acknowledge that we were unable to complete this experiment due to resource constraints. Nevertheless, we sought alternative ways to validate related aspects of iLLaMA's potential. Please allow us to elaborate further below.
>
> 1. **Resource Limitations in Training a CLIP-Like Model**
>    The LLaVA multimodal framework is built on a pre-trained CLIP [2] ViT-L/14 backbone. For a fair comparison, adapting iLLaMA to such a framework would require pre-training an iLLaMA-L/14 model in a manner similar to CLIP, followed by the multimodal fine-tuning process used in LLaVA. However, CLIP's pre-training relied on a massive dataset of 400M image-text pairs, which is not publicly available and whose scale and cost make it infeasible for most researchers, including ourselves, to replicate. This posed a significant barrier, and we regretfully acknowledge that we could not perform this experiment at the time of submission.
>
>    Despite these challenges, we recognize the importance of this evaluation and have explored alternative methods to approximate iLLaMA’s potential in cross-modal tasks.
>
> 2. **Indirect Evaluation of CLIP-Like Capabilities**
>    To compensate for our inability to directly test iLLaMA in a CLIP-like setting, we took inspiration from [3], which conducted detailed comparisons of ConvNeXt and ViT across various dimensions, such as model calibration, shape/texture bias, and transfer learning performance. Building on this idea, we compared iLLaMA to ConvNeXt [4] and ViT [5] along similar dimensions. These comparisons, which are detailed in Table.1, 2 and Table.3, provided strong evidence of iLLaMA's robustness and reliability.
>
>    Specifically, iLLaMA demonstrated comparable performance to ConvNeXt and ViT in terms of model calibration, shape/texture bias, and transfer learning capabilities. While these evaluations do not directly prove iLLaMA's suitability for a CLIP-like task, they offer valuable indirect evidence that suggests its potential as a competitive alternative. We hope this reassures readers of iLLaMA’s capability to align with and perhaps even surpass these established models in a unified multimodal framework.
>
>
>
>    Table 1: Calibration (↓) and shape-texture bias (↑) results.
>
>    | Evaluation         | ConvNeXt-B | DeiT3-B | iLLaMA-B |
>    | ------------------ | :--------: | :-----: | :------: |
>    | Calibration        |   0.0281   | 0.0415  |  0.0335  |
>    | Shape-Texture Bias |   33.30%   | 39.86%  |  41.45%  |
>
>    Table 2: CIFAR-10/100 results.
>
>    | Model    | CIFAR10 | CIFAR100 |
>    | -------- | :-----: | :------: |
>    | ViT-T    |  98.0   |   85.5   |
>    | iLLaMA-T |  97.9   |   85.5   |
>
> 3. **Future Directions and Community Collaboration**
>    We fully recognize that the above evidence is only an approximation and not a substitute for direct experiments. For this reason, we remain committed to pursuing this line of research. We are actively working to scale up resources and enable future experiments that directly evaluate iLLaMA in cross-modal settings. Additionally, should our paper gain broader attention, we hope that the open-sourcing of our code will encourage collaboration from the research community, enabling us and others to explore iLLaMA's potential in multimodal applications, such as CLIP and LLaVA, more comprehensively.
>
> In conclusion, while we regret our inability to directly demonstrate iLLaMA's cross-modal transferability at this time, we have taken meaningful steps to provide indirect evidence and lay the groundwork for future exploration. We sincerely hope this addresses your concerns and demonstrates our commitment to advancing this important research direction.
>
> Thank you once again for your invaluable feedback and for encouraging us to pursue this critical avenue.
>
>
>
> [1] Visual Instruction Tuning
>
> [2] Learning Transferable Visual Models From Natural Language Supervision
>
> [3] ConvNet vs Transformer, Supervised vs CLIP: Beyond ImageNet Accuracy
>
> [4] A ConvNet for the 2020s
>
> [5] An Image is Worth 16x16 Words: Transformers for Image Recognition at Scale

---

> > ### Author Response · Authors · 2024-11-25
> > **Response Part 3: Why We Need Softmask and Its Relationship to Our Design Intent**
> >
> > Thank you for raising this excellent question regarding the necessity of the softmask strategy. We are genuinely delighted to have the opportunity to address this, as the softmask strategy is indeed a critical component we designed to enhance the performance of vision Transformers operating in causal mode attention.
> >
> > The softmask strategy enforces a bi-directional attention mode during the early stages of training, gradually transitioning to causal mode attention following a configurable schedule. By inference time, the model fully adopts causal mode attention. We validated the effectiveness of this approach experimentally in **Table 1**, where softmask was shown to significantly improve the performance of iLLaMA. Furthermore, in **Table 2**, we demonstrated the generalizability of softmask across different scheduling strategies.
> >
> > Table 1: Ablation results of soft mask.
> >
> > | Model       |  Training Loss  |  Testing Loss   | ImageNet top-1 Accuracy (%) |
> > | :---------- | :-------------: | :-------------: | :-------------------------: |
> > | Tiny        |      2.990      |      1.121      |            74.3             |
> > | + Soft Mask | 2.955 (↓ 0.045) | 1.092 (↓ 0.029) |            75.0             |
> > | Base        |      2.868      |      0.843      |            81.3             |
> > | + Soft Mask | 2.828 (↓ 0.040) | 0.831 (↓ 0.012) |            81.6             |
> >
> > Table 2: Soft mask scheduling ablation.
> >
> > | Schedule    | Cutoff Epochs | Tiny | Base |
> > | ----------- | ------------- | ---- | ---- |
> > | no softmask | -             | 74.3 | 81.3 |
> > | linear      | 25            | 74.8 | 81.6 |
> > | linear      | 50            | 74.9 | 81.5 |
> > | linear      | 100           | 74.9 | 81.5 |
> > | constant    | 25            | 74.7 | 81.5 |
> > | constant    | 50            | 74.8 | 81.5 |
> >
> >
> >
> > While we understand that our results in **Tables 1 and 2** may not fully resolve all concerns, we would like to share the motivation behind the development of the softmask strategy. Initially, we observed that the PS [CLS] optimization technique allowed causal-attention-based Transformers to train successfully. However, their performance still lagged behind bi-directional attention-based Transformers. This performance gap inspired us to design a novel training strategy tailored to causal mode attention.
> >
> > The core idea of the softmask strategy is to transition *softly* from bi-directional attention during training to causal mode attention by inference, maintaining alignment with our design principle of a unified causal architecture. In essence, the strategy ensures that the model leverages the stability of bi-directional attention in the early training stages while gradually adapting to causal mode attention, thereby improving both training efficiency and final performance.
> >
> > We believe that the softmask strategy addresses a fundamental challenge in causal attention training while remaining consistent with our initial design goals. We hope this clarifies its purpose and aligns with your thoughtful observations.
> >
> > Thank you once again for your insightful comments, which have provided us with an opportunity to explain our design choices more thoroughly.

---

> ### Author Response · Authors · 2024-11-25
> **Response Part 4: About Computational Efficiency of Causal Mode Attention**
>
> Dear Reviewer,
>
> Thank you for your insightful question and for your meticulous observation regarding the computational complexity of causal attention compared to bi-directional attention, as detailed in Table 2. To facilitate our discussion, I have reproduced the relevant calculations below for clarity.
>
> Table 3:  Computational complexity results. causal mask reduces FLOPs required in the self-attention.
>
> | Type  | Bi-directional  | Causal                                       |
> | ----- | --------------- | -------------------------------------------- |
> | FLOPs | $4ND^2 + 2N^2D$ | $4ND^2 + N^2D + (\lfloor N^2/2 \rfloor+1) D$ |
>
> From the analysis in Table 3, we can see that the computational complexity of a single layer of causal attention reduces by approximately $N^2D- (\lfloor N^2/2 \rfloor+1)D$ compared to bi-directional attention. Assuming a model with $L$ layers, this results in a total reduction of $LN^2D- (\lfloor N^2/2 \rfloor+1)D$ FLOPs across the entire model.
>
> For example, in the case of the iLLaMA-T model with an input resolution of 224px, patch size of 16, and hidden dimension $D=192$, the reduction in computational cost can be approximated as:
>
> $\frac{N^2D- (\lfloor N^2/2 \rfloor+1)D}{4ND^2 + N^2D + (\lfloor N^2/2 \rfloor+1) D}\simeq 9.23\%$
>
> For models of this size, achieving nearly 10% savings in computation is particularly meaningful in resource-constrained deployment scenarios, such as edge devices or mobile platforms. In these settings, computational resources are precious, and optimizing the allocation of these resources to high-value operations is critical. The reduced computational overhead of causal mode attention creates room for other valuable computations, enhancing the overall efficiency and utility of the model.
>
> We hope this discussion highlights the practical advantages of causal attention, particularly for deployment on resource-limited platforms. Thank you again for this excellent observation, which has given us the chance to further elucidate this aspect of our work.
>
>
>
>
>
> We deeply appreciate the time and effort you have dedicated to reviewing our work. We sincerely hope that our detailed clarifications and discussions have addressed some of your concerns regarding the novelty, design choices, and performance aspects of iLLaMA. While we fully respect and understand your initial score, we kindly ask if you might reconsider your evaluation in light of the additional explanations provided. If our responses have helped resolve some of the doubts you initially had, we would be truly grateful if you could reflect this in your score. For instance, a revision from 3 to 5 would not only affirm the value of our work but also encourage us as researchers striving to contribute meaningfully to the field.
>
> Once again, we are humbled by your detailed and constructive feedback, which has already greatly enriched the quality of our work. Thank you for considering this request, and we remain open to further suggestions or concerns you may have.

---

> ### Author Response · Authors · 2024-11-26
> **Sincerely hope to have further discussion with you**
>
> Thank you again for your thoughtful and constructive feedback. Your comments have been invaluable in helping us clarify our work and address critical areas for improvement. We hope our responses have addressed your concerns, but we would be delighted to engage in further discussion if you have additional questions or suggestions. Your insights are deeply appreciated, and we look forward to any opportunity to learn from your expertise.
>
> Best,
>
> Authors

---

> ### Comment · Reviewer_6zNt · 2024-11-30
>
> First of all, sorry for the late response.
> Overall, my concerns have been partially addressed by the authors' response, but I still have doubts about whether this work provides sufficient value to the field.
> - While the authors offer a rationale for their design choices, I remain unconvinced because the response mainly emphasizes architectural alignment without clearly demonstrating additional value or novel outcomes.
> - They present indirect evidence to support iLLaMA's potential in cross-modal tasks, which is understandable given their constraints. However, the response does not fully satisfy my need for direct evidence of cross-modal transferability, which I believe is crucial. Cross-modal transfer is a significant weakness of this study, as noted by 3 reviewers, and it is even more necessary for research focused on a unified architecture. The indirect comparisons provided by the authors demonstrate the potential of iLLaMA as a backbone, but they fall short of proving its cross-modal transferability. I also think that testing cross-modality transferability doesn't necessarily require a CLIP-like or LLaVA-like setting.
>
> I appreciate the authors' detailed response and their constructive attitude. Consequently, I have raised my score to 5, but I am still inclined to reject the paper.

---

> ### Author Response · Authors · 2024-11-30
> **Heartfelt Gratitude for Your Kindness and Thoughtful Feedback**
>
> Dear Reviewer,
>
> We are truly surprised and deeply touched by your warm response and the thoughtful score adjustment. Please allow us to extend our sincerest gratitude and highest respect for your rigorous, responsible, and inclusive review process. Your recognition feels like a guiding star in the night, illuminating our journey as researchers and inspiring us to strive for better contributions. It has been an honor to have a reviewer like you, whose feedback is both critical and constructive.
>
> With this response, we aim to carefully address the concerns you have raised. While we may not be able to fully meet all your expectations, we will do our best to articulate the rigor, innovation, and value that we believe iLLaMA brings to the field.
>
>
>
> ##### **The Importance of Cross-Modal Validation**
>
> First and foremost, we completely agree with your emphasis on the importance of cross-modal tasks as a key benchmark for neural network architectures. Indeed, cross-modal capabilities are critical in evaluating the flexibility and utility of modern models. In this reply, we humbly wish to restate the value and contribution of iLLaMA to the research community, even though we acknowledge the absence of direct cross-modal transferability validation.
>
>
>
> ##### **iLLaMA’s Value as the First Step in Unified Architectures**
>
> We would like to share a perspective inspired by the development trajectory of some classical neural network architectures, such as Transformers. For example:
>
> 1. The Transformer architecture first demonstrated its effectiveness on visual tasks through ViT [1].
> 2. It was later extended to cross-modal tasks with CLIP [2].
> 3. Finally, its multimodal understanding capabilities were showcased through frameworks like LLaVA [3].
>
> This progression highlights that groundbreaking architectures often take a step-by-step validation approach rather than achieving everything within a single paper. The foundational "first step" of validating performance on pure visual tasks is crucial for enabling subsequent cross-modal and multimodal validations.
>
> In the context of our work, iLLaMA represents this critical "first step." By achieving a remarkable **86.0% top-1 accuracy on ImageNet-1K**, iLLaMA successfully demonstrates that a **decoder-only Transformer, fully based on causal attention**, can excel in vision tasks. This milestone was made possible through the innovations we introduced, particularly **PS [CLS]** and the **softmask strategy**, which enabled iLLaMA to overcome training collapse and achieve strong visual performance. We believe this foundational success is meaningful and sets the stage for further exploration in cross-modal tasks.
>
>
>
> ##### **Our Acknowledgment of Cross-Modal Transferability and Commitment to the Community**
>
> We wholeheartedly agree that cross-modal transferability is an essential aspect for evaluating unified architectures. As mentioned in our earlier responses, we regretfully acknowledge that due to resource constraints, we were unable to conduct comprehensive cross-modal experiments. However, we made two efforts to compensate for this limitation:
>
> 1. **Indirect Evaluations**: We evaluated iLLaMA’s **Model Calibration**, **Shape/Texture Bias**, and **transfer learning performance** to provide indirect evidence of its robustness and potential for cross-modal tasks. While these evaluations do not substitute for direct testing, they offer a strong foundation for understanding iLLaMA’s capabilities.
> 2. **Open-Sourcing Commitment**: To facilitate further validation, we are committed to making iLLaMA’s code open-source. We believe this will enable the community to build on our work, explore its cross-modal potential, and push it toward more impactful contributions in the field.
>
> We humbly hope that, as iLLaMA gains attention and adoption, the collective efforts of the research community will help advance its development into cross-modal and multimodal domains.
>
>
>
> ##### **A Humble Request**
>
> Finally, we deeply appreciate your kind feedback and the score adjustment to 5. If you feel that our response resolves even a small part of your concerns or provides additional clarity, we would be extremely grateful if you could consider readjusting the score. Even a minor difference would mean the world to us, as it reflects your acknowledgment of our efforts to address your concerns and improve our work.
>
> Thank you again for your time, thoughtfulness, and constructive guidance. It has been an honor to receive your detailed feedback, and we look forward to contributing more meaningfully to the field in the future.
>
>
>
> With heartfelt gratitude,
>
> Authors
>
>
>
> [1] An Image is Worth 16x16 Words: Transformers for Image Recognition at Scale
>
> [2] Learning Transferable Visual Models From Natural Language Supervision
>
> [3] Visual Instruction Tuning

---

> > ### Comment · Reviewer_6zNt · 2024-12-02
> >
> > The discussion on the paper seems sufficient, and now the judgment is up to the area chair. I acknowledge the value of this paper and the efforts of the authors, but in its current state, I believe there are various pros and cons. It seems appropriate for the area chair to make a decision after considering the discussions with other reviewers. I will maintain my score.

---

> ### Author Response · Authors · 2024-12-02
> **A Humble Request for Reconsideration**
>
> Dear Reviewer,
>
> Thank you for your thoughtful reply and for acknowledging both the value of our work and the efforts we have put into it. We truly appreciate your fair assessment of the pros and cons of iLLaMA, as well as your understanding of the importance of this research.
>
> We humbly and sincerely ask if you might reconsider the score. A higher score could help bring greater attention to iLLaMA, encouraging broader recognition of its contributions and enabling further exploration and improvement through community engagement. As a knowledgeable and experienced reviewer, we believe you value the importance of ensuring that promising work is not overlooked and that the strengths of a contribution are weighed more significantly than its minor shortcomings.
>
> We remain grateful for your thoughtful feedback and constructive evaluation, regardless of your final decision. Thank you once again for your time and effort.
>
> Sincerely,
>
> Authors

---

### Author Response · Authors · 2024-11-26
**General Response: Thanks, Contributions, and Revised Paper Updates**

Dear Area Chair and Reviewers,

We would like to express our heartfelt gratitude to all reviewers for their insightful and constructive feedback. Your comments have been instrumental in helping us refine and strengthen our work. The purpose of this letter is to thank the AC and all reviewers for the valuable time and effort you have devoted to evaluating our paper and to summarize the contributions of our work, as well as highlight the updates we have made in the revised version.



##### Contributions of Our Paper

1. **Unified Vision and Text Architecture**
   To harness the benefits of standardization in AI system architectures, we are committed to exploring a unified vision and text framework. This paper represents the first step in this direction, aiming to align the renowned decoder-only architecture of large language models (LLMs) with the vision domain. we explore several designs for adapting the LLaMA decoder as an image classifier, learning useful lessons along the way. iLLaMA fully aligns with LLaMA in terms of architecture, providing a unified approach for vision and text tasks.
2. **Addressing Attention Collapse**
   We identify the **attention collapse issue** when applying causal-mode attention and introduce the **post-sequence [CLS] (PS [CLS])** technique along with a **softmask** strategy to effectively address this challenge and improve model training behavior.
3. **Extensive Experiments**
   Comprehensive experiments on **ImageNet**, **transfer learning** tasks, and practical evaluations such as **quantization compatibility**, **model calibration**, and **shape-texture bias** demonstrate that iLLaMA can serve as an efficient and reliable alternative to ViTs for image feature extraction.



##### Updates in the Revised Version

1. **Updated Table 6**
   We have updated results in Table 6 and the relevant clarification for vision architectures based on Mamba and xLSTM as comparisons, addressing reviewer feedback and enhancing the comprehensiveness of our evaluation.
2. **Clarifications in Appendix K**
   We have updated clarifications on attention map visualizations have been included in Appendix K, addressing reviewer concerns and providing a more detailed explanation of our observations.



The revised version of our paper has incorporated these updates, and we hope it meets the expectations of the AC and reviewers. Once again, we deeply appreciate the time and effort you have invested in reviewing our submission. It is our sincere hope that iLLaMA’s contributions to the field will justify the dedication and care you have shown throughout this process. This remains our ultimate goal and source of motivation.

Thank you again for your invaluable support and feedback.

---

### Meta-Review · Area_Chair_VtHT · 2024-12-17

**Metareview:**

This paper presents an interesting idea that makes use of the architecture of the LLaMA model (a language model) as a vision transformer. For this purpose, the LLaMA architecture is step-by-step adjusted to fit the vision data (see Figure 1, the right-hand side). While the final architecture obtains acceptable results on ImageNet-1K, the accuracy is still inferior to vision architectures (e.g. Swin transformer). Adding ImageNet-21K for pre-training improves accuracy, but the numbers are not fairly compared against recent models. In summary, the adapted model does not show a clear benefit in visual understanding. This paper also has other drawbacks, such as (1) the performance on downstream tasks is weak, (2) the idea is closely related to VisionLLaMA, a published ECCV'24 paper, and (3) the fitness of vision and language data -- note that the architecture still needs extensive visual pre-training, and loading language pre-trained models does not help.

The reviewers gave diverse scores (3/5/8/8). While The AC reads the paper and all discussions carefully, and tends to concur with the major concerns of the reviewers. The method is inspired by the need for unifying vision and language understanding, but it is doubtful whether the architecture needs to be unified in this way. Also, as Reviewer **6zNt** pointed out, some statements made by the authors are not convincing enough. All of the above makes the final conclusion of the paper (in particular, the necessity of designing and training such a model) doubtful. Summarizing the above, although the paper is interesting, the clear weakness makes it difficult for the AC to recommend acceptance.

**Additional Comments On Reviewer Discussion:**

The reviewers are diverse in their judgments. Even after the rebuttal, their recommendations did not converge. The positive reviewers mostly argue that the paper is interesting and that using the same architecture can facilitate the unification of vision and language, while the negative reviewers are doubtful whether vision-language unification can be achieved following this path (whether using the same architectures is important). The AC agrees with the negative reviewers more than the others.

---

### Decision · Program_Chairs · 2025-01-22

Reject